# ECO: Quantized Training without Full-Precision Master Weights

**Mahdi Nikdan** [1 2]  **Amir Zandieh** [1]  **Dan Alistarh** [2]  **Vahab Mirrokni** [1]

## Abstract

Quantization has significantly improved the compute and memory efficiency of Large Language Model (LLM) training. However, existing approaches still rely on accumulating their updates in high-precision: concretely, gradient updates must be applied to a high-precision weight buffer, known as *master weights*. This buffer introduces substantial memory overhead, particularly for Sparse Mixture of Experts (SMoE) models, where model parameters and optimizer states dominate memory usage. To address this, we introduce the Error-Compensating Optimizer (ECO), which eliminates master weights by applying updates directly to quantized parameters. ECO quantizes weights after each step and carefully injects the resulting quantization error into the optimizer momentum, forming an error-feedback loop with no additional memory. We prove that, under standard assumptions and a decaying learning rate, ECO converges to a constant-radius neighborhood of the optimum, while naive master-weight removal can incur an error that is inversely proportional to the learning rate. We show empirical results for pretraining small Transformers (30–800M), a Gemma-3 1B model, and a 2.1B parameter Sparse MoE model with FP8 quantization, and fine-tuning DeepSeek-MoE-16B in INT4 precision. Throughout, ECO matches baselines with master weights up to near-lossless accuracy, significantly shifting the static memory vs validation loss Pareto frontier.

## 1. Introduction

Scaling Large Language Model (LLM) training comes with substantial computational and memory costs. As models

[1]Google Research [2]Institute of Science and Technology Austria (ISTA). Correspondence to: Mahdi Nikdan <nikdan-mahdi@gmail.com>, Amir Zandieh <zandieh@google.com>, Dan Alistarh <dan.alistarh@ist.ac.at>.

*Proceedings of the 43rd International Conference on Machine Learning*, Seoul, South Korea. PMLR 306, 2026. Copyright 2026 by the author(s).

have grown from billions to trillions of parameters, training memory has become a central bottleneck. Low-precision training has therefore emerged as a practical direction: recent FP8 (Peng et al., 2023; Liu et al., 2024a), and even lower precision (Panferov et al., 2025) training methods can reduce activation memory and accelerate training while maintaining stable optimization.

Despite this progress, a key overhead in quantized training remains untouched: the presence of *master weights*. Most quantized and quantization-aware training pipelines still preserve a high-precision copy of the parameters (typically FP32) to accumulate gradient updates. This is largely because many updates are smaller than the discretization gap of low-precision formats: applying them directly to quantized weights can make updates vanish or incur large quantization noise. As a result, the model weight memory footprint often stays similar to the high-precision baseline, even when the forward and backward passes are heavily quantized. Even carefully engineered FP8 training systems explicitly retain high-precision accumulators for stability (Peng et al., 2023; Liu et al., 2024a). The issue is especially pronounced for Sparse Mixture of Experts (SMoE) models, where only a subset of parameters is active per token, yet *all* master weights must reside in memory.

More broadly, attempts to avoid high-precision accumulation either do not scale to LLM training (Lin et al., 2022) or have only been effective in narrow settings (Zhang et al., 2025). This leaves a clear gap: a general method that removes master weights without sacrificing convergence or introducing additional memory overhead. Eliminating master weights can yield memory savings comparable to quantizing optimizer states (e.g., momentum buffers), an approach that has been widely explored and is very popular (Dettmers et al., 2021).

In this work, we introduce the **E**rror-**C**ompensating **O**ptimizer (ECO), which enables accurate quantized training without full-precision master weights, and thus *zero* extra memory overhead. The key idea is the following: after updating each layer's parameters, we quantize the updated weights and inject the resulting quantization error into the optimizer's momentum buffer. This creates an error-feedback loop that *carries forward* the lost updates and compensates for them in subsequent steps, allowing updates

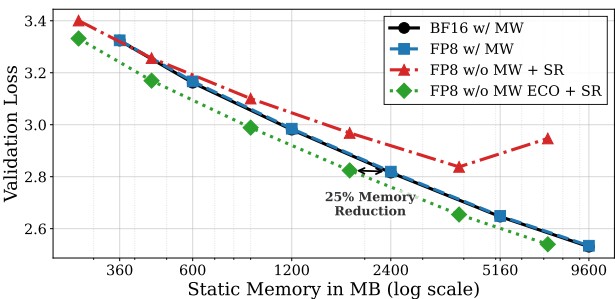

*Figure 1.* Static Memory Used vs Validation Loss comparing the standard BF16, FP8 with Master weights (FP8 w/ MW) baselines with standard stochastic rounding (FP8 w/o MW + SR) and ECO. ECO with stochastic rounding (SR) provides a significantly better Pareto frontier. Gradient accumulation is disabled in all cases.

to be applied directly to quantized parameters.

The resulting ECO iteration is simple to implement and requires no extra hyperparameter tuning. It further comes with theoretical guarantees. We study the convergence behavior of ECO applied to the SGD with momentum optimizer with momentum factor $\beta$. Under standard non-convex assumptions and a decaying learning rate, we prove that ECO converges to a constant-radius neighborhood of the true optimum. Moreover, this radius is only a $\frac{1}{1-\beta^2}$ factor worse than the best achievable bound when using master weights, where a nonzero error is unavoidable because the solution must lie on the quantization grid. We further construct a quadratic example showing that this bound is tight up a constant factor. In the same example, we show that naively removing master weights (without momentum error injection) yields a stationary error that scales inversely with the learning rate, and therefore diverges as the learning rate decays to zero.

We evaluate ECO with FP8 quantization across scaling law studies on small transformers (30M–800M parameters) (Panferov et al., 2025; Castro et al., 2025), pre-training a Gemma-3 1B (Gemma et al., 2025) and an SMoE 2.1B model, and fine-tuning a DeepSeek-MoE-16B model (Dai et al., 2024). Across settings, ECO nearly matches the validation loss of baselines that rely on master weights while significantly outperforming naive master weight removal. Furthermore ECO can reduce static memory usage by up to 25%, shifting the Pareto frontier between memory consumption and validation loss, as illustrated in Figure 1.

## 2. Related Work

**Quantized/Quantization-Aware Training.** Quantization-aware training (QAT) aims to enable low-precision inference by simulating quantization effects on weights and optionally activations during training (Esser et al., 2019; Choi et al., 2018; Panferov et al., 2025; Zhou et al., 2016; Jung et al., 2019; Wang et al., 2023b; Chen et al., 2025; Liu et al.,

2024b). Quantized training methods go further by quantizing the backward pass computation to accelerate training (Ashkboos et al., 2025; Castro et al., 2025; Tseng et al., 2025; Chmiel et al., 2025; Liu et al., 2024a). Post-training quantization (PTQ) methods such as Frantar et al. (2022); Lin et al. (2024); Ashkboos et al. (2024); Liu et al. (2024c); Sun et al. (2024) are computationally cheaper, but they typically incur larger accuracy degradation than QAT, especially at very low precision. Despite these advances, most QAT frameworks still rely on high-precision master weights to accumulate updates. Even recent QAT training systems such as FP8-LM (Peng et al., 2023), DeepSeek-V3 (Liu et al., 2024a), and Kimi-K2 (Team et al., 2025), who have rigorously tuned their quantization scheme, explicitly keep high-precision accumulators to maintain stability. In this context, ECO is complementary to existing QAT and quantized training methods: it targets the remaining dependence on master weights.

**Efforts Towards Low-Precision Accumulation.** Avoiding master weights has proven difficult outside restricted settings. FP8-LM reports that FP8 accumulation fails at large LLM scales (Peng et al., 2023). Lin et al. (2022) show that with careful gradient rescaling, INT8 accumulators can be stable for small convolutional networks that fit within 256KB of memory. APT (Huang et al., 2022) varies accumulator bit-width across layers for edge-device training. Collage (Yu et al., 2024) replaces FP32 with two BF16 accumulators due to a hardware constraint. Ozkara et al. (2025) argue that stochastic rounding is important for BF16 accumulation, and ELMO (Zhang et al., 2025) applies stochastic rounding to reduce the accumulator precision of the LLM head layer to BF16/FP8. Overall, there exists no general approach that enables sub-16-bit accumulation for large-scale LLM training, leaving an important gap that ECO addresses.

**Optimizer State Quantization.** A related line of work quantizes optimizer states (e.g., first and second moments) rather than model weights. In practice, the first moment is often more tolerant to quantization than the second. FP8-LM (Peng et al., 2023) reports that the first moment can be quantized to FP8 without difficulty. Other approaches quantize both moments to 8-bit (Dettmers et al., 2021; Fishman et al., 2024; Xi et al., 2024a), and Li et al. (2023) pushes this to 4-bit for both buffers. ECO targets a different bottleneck: the master-weight copy. This provides memory savings comparable to optimizer-state quantization, while remaining largely unexplored.

**Error Feedback.** Error feedback (EF) methods were developed to mitigate bias from compressed or quantized gradients, particularly in distributed optimization. They accumulate quantization residuals locally and add them back in

later steps, preserving the sum of updates over time (Seide et al., 2014; Tang et al., 2021; Wang et al., 2023a; Richtárik et al., 2021). Richtárik et al. (2021) provides a principled EF formulation and shows that it can match full-precision SGD convergence under appropriate assumptions. Directly applying EF to the master weight quantization requires storing an error buffer, which conflicts with memory reduction goals when training at scale. ECO instead reuses the optimizer momentum buffer to store quantization error, achieving error feedback without any extra memory.

## 3. Method

In this section, we start by introducing the notation and covering relevant background. We then describe our main method ECO. Finally, we present our theoretical results which analyze the convergence of ECO.

### 3.1. Notation and Background

**Notation.** Throughout this section, we denote the model parameters by $\boldsymbol{\theta}$ and their corresponding gradients by $\mathbf{g}$. The optimizer's first and second momentum buffers are represented by $\mathbf{m}$ and $\mathbf{v}$, respectively, with their corresponding coefficients denoted by $\beta_1$ and $\beta_2$ (or just $\beta$ in case of SGD). We denote the quantization as $q(\cdot)$, and $\mathbf{e}$ represents the quantization error (e.g., $\mathbf{e}_{\boldsymbol{\theta}} = \boldsymbol{\theta} - q(\boldsymbol{\theta})$), and $\eta$ is the learning rate.

**Quantization.** Quantization is the process of mapping continuous or high-precision values to a low-precision representation, primarily to reduce memory usage and enhance arithmetic throughput. This process typically involves an affine transformation (scaling by $s$ and shifting by $z$) to project the original values into the target range, followed by a rounding function that maps each value to the nearest grid point.

More formally, a high-precision vector $\mathbf{x}$ is quantized to a low-precision vector $\mathbf{y}$ using the formula $\mathbf{y} = round(\frac{\mathbf{x}-z}{s})$. The original values can then be approximated using $\hat{\mathbf{x}} = s\mathbf{y} + z$. Thus, the fully reconstructed vector $\hat{\mathbf{x}}_{z,s}$ is calculated as:

$$\hat{\mathbf{x}}_{z,s} = s \cdot round(\frac{\mathbf{x}-z}{s}) + z \cdot \qquad (1)$$

Assuming the largest quantized value representable by the quantization format is $\rho$, then a standard choice for the scaling factor is $s = \max|\mathbf{x}|/\rho$, which prevents overflow. It is also common to fix the zero-point $z = 0$, particularly for tensors in LLM training that are often near zero-mean. Therefore, for simplicity, when $z$ and $s$ are not explicitly mentioned, we assume this symmetric scheme, i.e., $\hat{\mathbf{x}} = q(\mathbf{x}) = \frac{\max|\mathbf{x}|}{\rho} \cdot round(\frac{\rho \mathbf{x}}{\max|\mathbf{x}|})$.

Quantization schemes can be categorized in several ways. One key distinction is their granularity, which defines which parts of an input tensor share the same quantization parameters (i.e., zero-point $z$ and scale $s$). For example, in row-wise quantization, an independent $z$ and $s$ are computed and applied to each row of an input matrix. Other methods exists, such as 1D or 2D group-wise quantization, where blocks or groups of elements within the tensor share quantization parameters (Liu et al., 2024a; Xi et al., 2024b; Rouhani et al., 2023).

Another categorization stems from the rounding function. A standard choice is round-to-nearest, which deterministically maps each value to its closest grid point. Alternatively, stochastic rounding maps a value to one of the two nearest grid points, where the probability of selecting either point is proportional to the distance to the other point. Round-to-nearest minimizes the magnitude of the error, while stochastic rounding results in an unbiased estimator.

**Quantization-Aware Training with Master Weights.** Most quantized LLM training pipelines keep high-precision master weights (typically FP32) as the update accumulator. At each step, the master weights are quantized to obtain low-precision weights used for the forward/backward pass, while gradients and optimizer updates are accumulated in the high-precision copy. This stabilizes training by preserving small updates, but it substantially limits the weight-memory savings of quantization: the full master-weight buffer must remain on memory throughout training.

### 3.2. ECO

The high-level idea of ECO is to *inject* the quantization error from the current step into the optimizer's momentum buffer. This mechanism ensures that the error from the current step is *carried over* and incorporated into the parameter update of the subsequent step, effectively creating an error feedback loop. Algorithm 1 provides a general overview, while Algorithm 2 and Algorithm 3 detail the error injection process for the SGD with Momentum (SGDM) and Adam optimizers, respectively.

**SGDM.** ECO applies SGDM updates directly to the quantized weights. Concretely, at step $t$ with low-precision parameters $\hat{\boldsymbol{\theta}}_t$, it forms a temporary iterate $\tilde{\boldsymbol{\theta}}_{t+1} = \hat{\boldsymbol{\theta}}_t + \mathbf{u}_t$ (where $\mathbf{u}_t$ is the SGDM update, dominated by momentum), quantizes it to obtain $\hat{\boldsymbol{\theta}}_{t+1} = q(\tilde{\boldsymbol{\theta}}_{t+1})$, and defines the quantization error $\mathbf{e}_{t+1} := \tilde{\boldsymbol{\theta}}_{t+1} - \hat{\boldsymbol{\theta}}_{t+1}$. ECO then injects this error into the momentum buffer so that the update lost due to quantization is carried forward and recovered in later steps.

We prove in Appendix A that, if the errors are injected into momentum as

$$\mathbf{m} \leftarrow \mathbf{m} + \frac{1}{\eta}\mathbf{e}_t - \frac{1}{\eta\beta}\mathbf{e}_{t+1},$$

then the resulting optimization trajectory is *identical* to

SGDM with master weights. The difficulty is that this exact rule is not memory-efficient: while $\mathbf{e}_{t+1}$ is available on-the-fly from the current quantization, the previous-step residual $\mathbf{e}_t$ must be stored, which reintroduces a persistent buffer.

We tackle this issue by a heuristic observation: $\mathbf{e}_{t+1}$ and $\mathbf{e}_t$ are typically close. Intuitively, assuming a fixed scale parameter, $\hat{\boldsymbol{\theta}}_t$ is already on-grid, so moving to the next iterate only quantizes the *increment* $\mathbf{u}_t$, i.e., $q(\hat{\boldsymbol{\theta}}_t + \mathbf{u}_t) = \hat{\boldsymbol{\theta}}_t + q(\mathbf{u}_t)$. Since $\mathbf{u}_t$ is dominated by momentum, it changes slowly from one step to the next, which in turn makes the induced quantization errors $\mathbf{e}_{t+1}$ and $\mathbf{e}_t$ close. We also validate this empirically in Section 4.2. We therefore substitute $\mathbf{e}_t \approx \mathbf{e}_{t+1}$, yielding the memory-free injection rule

$$\mathbf{m} \leftarrow \mathbf{m} + \frac{1}{\eta}\left(1 - \frac{1}{\beta}\right)\mathbf{e}_{t+1},$$

which removes the need for either master weights or a stored error buffer. See Algorithm 2 for more details. Notably, we use this heuristic only to motivate the injection rule; later in this section, we provide a rigorous theoretical analysis of the resulting memory-efficient form.

**Adam.** We treat Adam in the same way as SGDM, except that Adam applies an *adaptive*, element-wise learning rate. Adam's parameter update can be written in the form

$$\boldsymbol{\theta}_{t+1} = \boldsymbol{\theta}_t - \eta\frac{\frac{\mathbf{m}_{t+1}}{1-\beta_1^t}}{\sqrt{\frac{\mathbf{v}_{t+1}}{1-\beta_2^t}} + \epsilon},$$

where $\mathbf{m}_{t+1}$ and $\mathbf{v}_{t+1}$ are the first and second momentum buffers after incorporating the gradient at step $t$, and $\epsilon$ prevents division by zero. We identify the element-wise adaptive step size as

$$\boldsymbol{\eta}_t := \frac{\eta}{(1-\beta_1^t)(\sqrt{\frac{\mathbf{v}_{t+1}}{1-\beta_2^t}} + \epsilon)}.$$

With this formulation, ECO's injection differs from the SGDM case only by replacing the scalar learning rate with Adam's element-wise effective step size. See Algorithm 3.

---

**Algorithm 1** Quantized Training Step $t$ with ECO

---

**Require:** Quantized parameters $\hat{\boldsymbol{\theta}}_t$
**Require:** Optimizer state $\hat{\mathbf{s}}_t$, hyperparameters $H$
**Require:** Optimizer step function: *OPTIM_STEP*
**Require:** ECO quantization function: *ECO_QUANTIZE*
 1: $\tilde{\boldsymbol{\theta}}_{t+1}, \tilde{\mathbf{s}}_{t+1} \leftarrow OPTIM\_STEP(\hat{\boldsymbol{\theta}}_t, \hat{\mathbf{s}}_t, H)$
 2: $\hat{\boldsymbol{\theta}}_{t+1}, \hat{\mathbf{s}}_{t+1} \leftarrow ECO\_QUANTIZE(\tilde{\boldsymbol{\theta}}_{t+1}, \tilde{\mathbf{s}}_{t+1}, H)$
 3: **return** $\hat{\boldsymbol{\theta}}_{t+1}, \hat{\mathbf{s}}_{t+1}$

---

**3.3. Convergence Analysis**

This section presents the convergence analysis for the SGDM variant of the ECO optimizer. By constructing a

---

**Algorithm 2** *ECO_QUANTIZE* for SGD with Momentum

---

**Require:** High-precision parameters $\tilde{\boldsymbol{\theta}}_{t+1}$
**Require:** Optimizer state $\tilde{\mathbf{s}}_{t+1}$, hyperparameter $H$
 1: $\hat{\boldsymbol{\theta}}_{t+1} \leftarrow q(\tilde{\boldsymbol{\theta}}_{t+1})$         ▷ quantize the weights
 2: $\mathbf{e}_{t+1} \leftarrow \tilde{\boldsymbol{\theta}}_{t+1} - \hat{\boldsymbol{\theta}}_{t+1}$ ▷ compute the quantization error
 3: $\{\tilde{\mathbf{m}}_{t+1}\} \leftarrow \tilde{\mathbf{s}}_{t+1}$      ▷ read momentum buffer from the optimizer state
 4: $\{\eta, \beta\} \leftarrow H$          ▷ read SGDM hyperparameters
 5: $\hat{\mathbf{m}}_{t+1} \leftarrow \tilde{\mathbf{m}}_{t+1} + \frac{1}{\eta}(1 - \frac{1}{\beta})\mathbf{e}_{t+1}$      ▷ inject the quantization error into momentum
 6: **return** $\hat{\boldsymbol{\theta}}_{t+1}, \{\hat{\mathbf{m}}_{t+1}\}$

---

**Algorithm 3** *ECO_QUANTIZE* for Adam

---

**Require:** High-precision parameters $\tilde{\boldsymbol{\theta}}_{t+1}$
**Require:** Optimizer state $\tilde{\mathbf{s}}_{t+1}$, hyperparameter $H$
 1: $\hat{\boldsymbol{\theta}}_{t+1} \leftarrow q(\tilde{\boldsymbol{\theta}}_{t+1})$        ▷ quantize the weights
 2: $\mathbf{e}_{t+1} \leftarrow \tilde{\boldsymbol{\theta}}_{t+1} - \hat{\boldsymbol{\theta}}_{t+1}$ ▷ compute the quantization error
 3: $\{\tilde{\mathbf{m}}_{t+1}, \mathbf{v}_{t+1}\} \leftarrow \tilde{\mathbf{s}}_{t+1}$ ▷ read momentum buffers from the optimizer state
 4: $\{\eta, \beta_1, \beta_2, \epsilon\} \leftarrow H$     ▷ read Adam hyperparameters
 5: $\hat{\mathbf{m}}_{t+1} \leftarrow \tilde{\mathbf{m}}_{t+1} + \frac{1-\beta_1^t}{\eta}(1 - \frac{1}{\beta_1})(\sqrt{\frac{\mathbf{v}_{t+1}}{1-\beta_2^t}} + \epsilon) \odot \mathbf{e}_{t+1}$      ▷ inject the quantization error into momentum
 6: **return** $\hat{\boldsymbol{\theta}}_{t+1}, \{\hat{\mathbf{m}}_{t+1}, \mathbf{v}_{t+1}\}$

---

virtual sequence, we prove that the algorithm converges to a near stationary point. All proofs are given in Appendix B.

3.3.1. SETUP AND ALGORITHM

We consider the optimization problem $\min_{\boldsymbol{\theta}\in\mathbb{R}^d} f(\boldsymbol{\theta})$, where $f$ is $L$-smooth and bounded below by $f^*$.

The ECO Optimizer updates are expanded as follows:

$$\tilde{\mathbf{m}}_{t+1} = \beta\hat{\mathbf{m}}_t + (1 - \beta)\nabla f(\hat{\boldsymbol{\theta}}_t) \tag{2}$$

$$\tilde{\boldsymbol{\theta}}_{t+1} = \hat{\boldsymbol{\theta}}_t - \eta\tilde{\mathbf{m}}_{t+1} \tag{3}$$

$$\hat{\boldsymbol{\theta}}_{t+1} = q(\tilde{\boldsymbol{\theta}}_{t+1}) \tag{4}$$

$$\mathbf{e}_{t+1} = \tilde{\boldsymbol{\theta}}_{t+1} - \hat{\boldsymbol{\theta}}_{t+1} \tag{5}$$

$$\hat{\mathbf{m}}_{t+1} = \tilde{\mathbf{m}}_{t+1} + \alpha\mathbf{e}_{t+1} \tag{6}$$

where $\eta$ is the learning rate, $\beta \in [0, 1)$ is the momentum parameter, and the error injection strength is set to:

$$\alpha = \frac{1}{\eta}\left(1 - \frac{1}{\beta}\right). \tag{7}$$

3.3.2. ASSUMPTIONS

We rely on the following standard assumptions for non-convex optimization analysis.

**Assumption 3.1** (L-Smoothness)**.** The function $f$ is $L$-smooth, i.e., $\|\nabla f(x) - \nabla f(y)\| \le L\|x - y\|$ for all $x, y$.

**Assumption 3.2** (Unbiased Quantization with Bounded Error Variance). The quantization error is zero-mean with bounded variance $\sigma^2$: $\mathbb{E}[\mathbf{e}_t] = 0$ and $\mathbb{E}[\|\mathbf{e}_t\|^2] \leq \sigma^2$.

**Assumption 3.3** (Bounded Gradient). There exists $G > 0$ such that $\|\nabla f(\boldsymbol{\theta})\| \leq G$ for all $\boldsymbol{\theta}$.

### 3.3.3. VIRTUAL SEQUENCE ANALYSIS

Following the methodology of Richtárik et al. (2021), we construct a "virtual sequence" $\boldsymbol{\theta}_t$.

**Definition 3.4** (Virtual Sequence). Define the virtual sequence $\boldsymbol{\theta}_t$ as:

$$\boldsymbol{\theta}_t := \hat{\boldsymbol{\theta}}_t - \frac{\eta\beta}{1-\beta}\hat{\mathbf{m}}_t. \tag{8}$$

**Lemma 3.5** (Virtual Sequence Dynamics). *The virtual sequence $\boldsymbol{\theta}_t$ evolves as:*

$$\boldsymbol{\theta}_{t+1} = \boldsymbol{\theta}_t - \eta\nabla f(\hat{\boldsymbol{\theta}}_t), \tag{9}$$

This lemma demonstrates that by tracking this specific combination of weights and momentum, we can analyze the ECO trajectory as a standard gradient descent process on the loss surface.

### 3.3.4. DESCENT AND MOMENTUM BOUNDS

We derive a descent inequality for the virtual sequence and bound the momentum term which accumulates the quantization error.

**Lemma 3.6** (Descent Lemma). *Let $C = \frac{\eta\beta}{1-\beta}$. For $\eta \leq \frac{1}{2L}$, the virtual sequence satisfies:*

$$f(\boldsymbol{\theta}_{t+1}) \leq f(\boldsymbol{\theta}_t) - \frac{\eta}{4}\left\|\nabla f(\hat{\boldsymbol{\theta}}_t)\right\|_2^2 + \frac{\eta L^2 C^2}{2}\|\hat{\mathbf{m}}_t\|_2^2 \tag{10}$$

This allows us to control the dynamics of the optimization trajectory.

**Lemma 3.7** (Bounded Momentum). *Under the assumptions, the squared norm of the momentum $\hat{\mathbf{m}}_t$ is bounded in expectation by a constant $M^2$. Specifically, for all $t$:*

$$\mathbb{E}[\|\hat{\mathbf{m}}_t\|^2] \leq M^2 := 2G^2 + \frac{2\alpha^2\sigma^2}{1-\beta^2}. \tag{11}$$

This ensures that the quantization error injected into the momentum buffer does not explode, keeping the optimization stable.

### 3.3.5. CONVERGENCE THEOREM

**Theorem 3.8** (Convergence Rate). *For $\eta \leq \frac{1}{2L}$, the ECO optimizer converges to a neighborhood:*

$$\min_{t\in\{0,...,T-1\}} \mathbb{E}\left[\|\nabla f(\hat{\boldsymbol{\theta}}_t)\|^2\right] \leq \frac{4(f(\boldsymbol{\theta}_0) - f^*)}{\eta T} + \sigma_{quant}^2, \tag{12}$$

*where the quantization noise floor $\sigma_{quant}^2$ is given by:*

$$\sigma_{quant}^2 = \frac{4\eta^2\beta^2 L^2 G^2}{(1-\beta)^2} + \frac{4L^2\sigma^2}{1-\beta^2} \tag{13}$$

**Discussion on Decaying Learning Rate:** As $\eta \to 0$, the noise floor $\sigma_{\text{quant}}^2$ becomes:

$$\lim_{\eta\to 0} \sigma_{\text{quant}}^2 = \frac{4L^2\sigma^2}{1-\beta^2} \tag{14}$$

While the noise floor persists even as the learning rate vanishes, we show in the next subsection that this noise floor is tight up to the constant $4$. Additionally, we note that even with master weights, since the final solution must lie on the quantization grid, a noise floor of $L^2\sigma^2$ is unavoidable.

### 3.3.6. DETERMINISTIC ROUNDING

We now provide a similar study where deterministic round-to-nearest is used instead of stochastic rounding. In this case, the zero-mean error assumption (Assumption 3.2) is violated. We instead assume a bounded deterministic error $\|\mathbf{e}_t\| \leq \delta$ for all $t$.

**Lemma 3.9** (Deterministic Momentum Bound). *Under the deterministic error assumption $\|\mathbf{e}_t\| \leq \delta$ and bounded gradients $\|\nabla f(\boldsymbol{\theta})\| \leq G$, the norm of the injected momentum buffer in ECO is uniformly bounded for all $t$:*

$$\|\hat{\mathbf{m}}_t\| \leq M_{det} := G + \frac{|\alpha|\delta}{1-\beta}. \tag{15}$$

**Theorem 3.10** (Deterministic Convergence). *For $\eta \leq \frac{1}{2L}$, the ECO optimizer with deterministic rounding converges to a neighborhood of the optimum:*

$$\min_{t<T} \|\nabla f(\hat{\boldsymbol{\theta}}_t)\|^2 \leq \frac{4(f(\boldsymbol{\theta}_0) - f^*)}{\eta T} + \Gamma_{quant}^2 \tag{16}$$

*where the deterministic noise floor is defined as:*

$$\Gamma_{quant}^2 = 2L^2 C^2 M_{det}^2 = \frac{2L^2\eta^2\beta^2}{(1-\beta)^2}\left(G + \frac{|\alpha|\delta}{1-\beta}\right)^2. \tag{17}$$

**Comparison of Noise Floors.** It is instructive to compare the noise floor of the stochastic case ($\sigma_{\text{quant}}^2$) and the deterministic case ($\Gamma_{\text{quant}}^2$) as the learning rate $\eta \to 0$. In the stochastic case, the noise floor remains constant at $\mathcal{O}(L^2\sigma^2/(1-\beta^2))$. In the deterministic case, substituting $|\alpha| = (1-\beta)/\eta\beta$ results in a floor of $\mathcal{O}(L^2\delta^2/(1-\beta)^2)$. Assuming $\sigma \approx \delta$, the deterministic bound is significantly larger due to the $(1-\beta)^{-2}$ dependence, reflecting the fact that systematic biases in quantization are harder for the momentum buffer to "average out" than zero-mean noise.

## 3.4. Lower-Bound on Worst-Case Behavior

We analyze the optimization dynamics on a one-dimensional quadratic objective $f(x) = \frac{L}{2}x^2$ with $L > 0$. The gradient is $\nabla f(x) = Lx$. We assume a stochastic quantization model where the quantized value $\hat{x} = q(x)$ satisfies $\hat{x} = x + \xi$, with $\xi$ being zero-mean noise independent of $x$ and $\mathbb{E}[\xi^2] = \sigma^2$. We examine the expected squared gradient norm of the stationary *quantized* parameters, defined as $\mathcal{L} = \lim_{t \to \infty} \mathbb{E}[(\nabla f(\hat{x}_t))^2]$, in the limit as the learning rate $\eta \to 0$. The results are summarized below, while the formal derivations are deferred to Appendix C.

**SGDM with Master Weights.** In this standard setting, the master weights evolve in high precision, but the gradient is computed using the quantized weights. Master weights allow the underlying parameter to converge to the true optimum. However, the quantized weights are $\xi$ away from the master weights. Consequently, the error is dominated by the quantization resolution:

$$\lim_{\eta \to 0} \mathcal{L}_{\text{MW}} = L^2 \sigma^2. \tag{18}$$

**Naive Master Weight Removal.** When master weights are removed, the update is applied directly to the quantized parameter: $\hat{x}_{t+1} = q(\hat{x}_t - \eta m_{t+1})$. This process reaches a stationary distribution, however, the variance is inversely proportional to the learning rate:

$$\mathcal{L}_{\text{Naive}} \propto \frac{1}{\eta} \xrightarrow{\eta \to 0} \infty. \tag{19}$$

This confirms that without error compensation, one cannot achieve high accuracy by annealing the learning rate.

**ECO.** ECO stabilizes the master-weight-free training by injecting quantization noise into the momentum buffer. In the limit of small learning rates, the process converges to a stationary distribution determined by the noise accumulation in the momentum term:

$$\lim_{\eta \to 0} \mathcal{L}_{\text{ECO}} = \frac{L^2 \sigma^2}{1 - \beta^2}. \tag{20}$$

This shows that ECO prevents the $1/\eta$ explosion seen in the naive case. Additionally, this verifies that the noise floor in in Equation (14) is tight up to a factor of 4.

# 4. Experiments

## 4.1. Baselines

We evaluate the following baselines that use high-precision accumulation.

- **FP32 accumulation with BF16 computation (BF16 w/ MW)**: This configuration serves as the reference baseline. Training is performed using FP32 master weights, while operands are cast to BF16 prior to each matrix multiplication to improve efficiency. This setup follows standard automatic mixed-precision training (Micikevicius et al., 2017) and provides an upper bound on achievable performance.

- **FP32 accumulation with FP8 round-to-nearest forward pass (FP8 w/ MW + RTN)**: This quantization-aware training (QAT) baseline quantizes both weights and activations to the FP8 E4M3 format during the forward pass using round-to-nearest. Row-wise scaling is applied, with each scale set to the maximum absolute value in the corresponding row. Prior work has shown that this approach is largely lossless (Liu et al., 2024a; Peng et al., 2023).

- **FP32 accumulation with FP8 stochastic rounding forward pass (FP8 w/ MW + SR)**: This baseline is identical to the previous one, except that weights are quantized using stochastic rounding. Activations remain quantized with round-to-nearest.

The baselines above maintain FP32 master weights and therefore establish upper bounds for the following methods, which eliminate master weight storage.

- **FP8 accumulation and forward pass with round-to-nearest (FP8 w/o MW + RTN)**: This baseline provides a direct comparison to ECO. No high-precision master weights are stored. After each parameter update, weights are quantized to FP8 using round-to-nearest. Activations are also quantized to FP8.

- **FP8 accumulation and forward pass with stochastic rounding (FP8 w/o MW + SR)**: This method mirrors the previous baseline, but applies stochastic rounding to the weights. Activations are still quantized using round-to-nearest. This corresponds to the approach suggested by Ozkara et al. (2025).

- **FP8 accumulation and forward pass with round-to-nearest and ECO (FP8 w/o MW ECO + RTN)**: In addition to removing master weights and applying round-to-nearest quantization to both weights and activations, this method incorporates our momentum injection mechanism to mitigate quantization error.

- **FP8 accumulation and forward pass with stochastic rounding and ECO (FP8 w/o MW ECO + SR)**: This variant is identical to the previous method, but uses stochastic rounding for weight quantization.

## 4.2. Scaling Law Experiments

**Setting.** We evaluate ECO using a pre-training scaling study, following Panferov et al. (2025). We train models

*Table 1.* Validation loss comparison across model sizes 30-800M, with "dvg" denoting divergence. *"N/A": one entry is unavailable due to data loss.

| Model Size | 30M | 50M | 100M | 200M | 430M | 800M |
|---|---|---|---|---|---|---|
| BF16 w/ MW | 3.3238 | 3.1616 | 2.9811 | 2.8157 | 2.6464 | 2.5306 |
| FP8 w/ MW + RTN | 3.3248 | 3.1668 | 2.9846 | 2.8194 | 2.6490 | 2.5343 |
| FP8 w/ MW + SR | 3.3309 | 3.1719 | 2.9884 | 2.8231 | 2.6500 | N/A* |
| FP8 w/o MW + RTN | dvg | dvg | dvg | dvg | dvg | dvg |
| FP8 w/o MW + SR | 3.4008 | 3.2563 | 3.1006 | 2.9684 | 2.8378 | 2.9471 |
| FP8 w/o MW ECO + RTN | 3.3640 | 3.1862 | 3.0025 | 2.8776 | 2.7237 | 2.6046 |
| FP8 w/o MW ECO + SR | **3.3317** | **3.1695** | **2.9888** | **2.8241** | **2.6544** | **2.5399** |

with sizes of 30M, 50M, 100M, 200M, 430M, and 800M parameters. For a model with $N$ parameters, training is performed on $100N$ tokens from the C4 dataset (Raffel et al., 2020), corresponding to $5\times$ the Chinchilla-optimal token count (Hoffmann et al., 2022). We use the T5 tokenizer (Raffel et al., 2020; Kudo & Richardson, 2018). Both the batch size and sequence length are fixed to 512. We use the AdamW optimizer with $(\beta_1, \beta_2, \epsilon) = (0.9, 0.98, 10^{-9})$. The learning rate is linearly warmed up from $0.01\times$ the peak value to the peak over the first $10\%$ of training, followed by cosine decay to $0.1\times$ the peak. We apply a weight decay of 0.1 and gradient clipping with a norm of 1.0. Refer to Panferov et al. (2025) for more details on the hyperparameters. For quantized runs, we apply the method only to the linear layers within transformer blocks, excluding the embedding and output layers.

**Results.**   Table 1 reports the final validation loss achieved by each method. The results show that ECO substantially improves over naive removal of master weights. When stochastic rounding is used, ECO nearly recovers the performance of methods that retain master weights. As expected, the gains are smaller with round-to-nearest quantization, since it introduces bias into the momentum buffer.

**Memory and Runtime.**   In addition, Figure 1 shows that ECO establishes a new static memory–loss Pareto frontier, offering significantly lower memory usage for a given validation loss. Regarding runtime, the injection is a simple element-wise operation and adds negligible overhead.

**Study on the Similarity of Consecutive Errors.**   We repeat the 30M experiment with master weights and round-to-nearest (RTN), and measure the similarity between consecutive quantization errors. Specifically, we track the relative norm $\frac{\|\mathbf{e}_{t+1}\|_2}{\|\mathbf{e}_t\|_2}$ and the cosine similarity between $\mathbf{e}_t$ and $\mathbf{e}_{t+1}$ throughout training. Figure 2 reports both metrics. The relative norm remains close to 1 during training, indicating that $\|\mathbf{e}_t\|_2$ varies slowly over time, and the cosine similarity stays consistently high, indicating strong alignment between

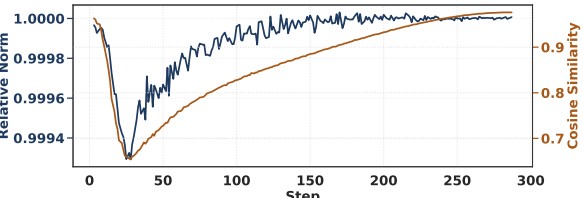

*Figure 2.* Similarity of consecutive quantization errors. Left: relative norm $\|\mathbf{e}_{t+1}\|_2 / \|\mathbf{e}_t\|_2$. Right: cosine similarity between $\mathbf{e}_t$ and $\mathbf{e}_{t+1}$.

consecutive errors. The observed trend follows the learning-rate schedule: larger learning rates lead to larger differences between consecutive errors, while these differences diminish as the learning rate decays.

### 4.3. Gemma 3 1B Pre-training

**Setting.**   We pre-train the Gemma 3 1B model (Gemma et al., 2025) from scratch on 40B tokens from the C4 dataset (Raffel et al., 2020). The batch size is 256 and the sequence length is 512. We use the publicly available Gemma 3 tokenizer. Training uses the AdamW optimizer with the same hyperparameters as in the scaling law experiments. The learning rate peaks at $10^{-4}$, with a linear warmup from $10^{-6}$ over the first $10\%$ of training, followed by cosine decay to $10^{-5}$.

**Results.**   Figure 3 compares the final validation loss across methods. The results confirm the effectiveness of ECO, particularly when combined with stochastic rounding.

### 4.4. Mixture of Experts Pre-training

**Setting.**   We pre-train a sparse mixture-of-experts (SMoE) model with 2.1B total parameters. The model contains 32 experts, of which 4 are activated per token. It consists of 24 transformer layers, each with a hidden dimension of 576, an intermediate dimension of 2304, and 9 attention heads. Training uses $100\times$ the number of active parameters in tokens from the LM1B dataset (Chelba et al., 2013). We reuse

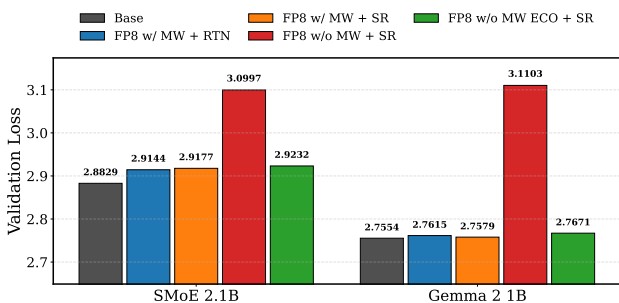

*Figure 3.* Gemma 3 1B and SMoE 2.1B validation loss comparison.

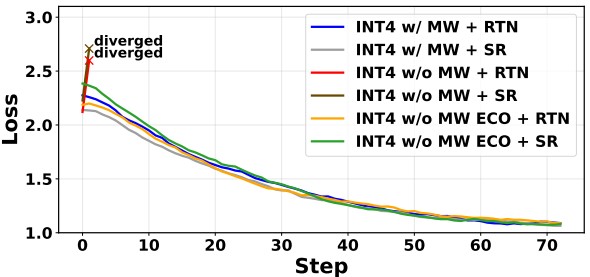

*Figure 4.* Smoothed training loss during fine-tuning of DeepSeek-MoE-16B-Base (Dai et al., 2024).

the T5 tokenizer (Raffel et al., 2020; Kudo & Richardson, 2018). Optimization is performed with AdamW, using a weight decay of $0.1$, and a learning rate that increases linearly from $2 \times 10^{-6}$ to $2 \times 10^{-5}$ over the first $1\%$ of training, followed by cosine decay back to $2 \times 10^{-6}$. The batch size is 256 and the sequence length is 512. For the quantized runs, we only quantize the expert linear layers.

**Results.** Figure 3 summarizes the final validation loss for each method. Consistent with prior experiments, ECO clearly outperforms naive master weight removal, while incurring only a minimal loss compared to approaches that retain master weights.

**Discussion on Memory.** Due to the SMoE model architecture, the memory required for activation storage is substantially smaller than that required for weights. With activation checkpointing enabled and no gradient accumulation, peak memory usage is dominated by master weights and optimizer states. Reducing master weight precision from FP32 to FP8 therefore lowers peak memory consumption from 12 bytes per parameter to 9, a reduction of approximately $25\%$.

### 4.5. DeepSeek-MoE-16B Fine-tuning

**Setting.** We apply ECO to tensor-wise INT4 weight-only QAT of DeepSeek-MoE-16B-Base (Dai et al., 2024). The model has 64 experts, with 8 active experts per token (approximately 2.8B parameters), including 2 shared experts. We fine-tune on the OpenAssistant-Guanaco dataset (Dettmers et al., 2023) for 3 epochs with sequence length 2048, using AdamW with micro-batch size 1 and gradient accumulation of 16. The learning rate is linearly warmed up from $2 \times 10^{-10}$ to $2 \times 10^{-5}$ over the first $3\%$ of training, then annealed to zero with a cosine schedule. We apply gradient clipping with threshold 1 and use no weight decay.

**Results.** Figure 4 compares training loss across methods. Naive master-weight removal diverges under both round-to-nearest (RTN) and stochastic rounding (SR), whereas ECO matches the master-weight baseline in both cases. In addition, Table 2 reports zero-shot accuracy on standard

benchmarks, where ECO similarly recovers the performance of the master-weight models.

*Table 2.* Fine-tuned DeepSeek-MoE-16B zero-shot benchmarks. We omit naive master weight removal baselines because training diverged in those settings. ECO matches the master-weight baselines, demonstrating lossless accuracy while requiring significantly less memory.

| Method | ARC-C | ARC-E | GSM8K | HellaSwag | PIQA | MMLU |
|---|---|---|---|---|---|---|
| Base | 47.53 | **73.06** | 16.15 | 77.34 | 80.36 | 37.64 |
| INT4 w/ MW + RTN | 48.29 | 71.38 | **16.68** | 78.76 | 80.69 | 37.87 |
| INT4 w/ MW + SR | 48.55 | 71.13 | 16.15 | 78.78 | 80.90 | 38.57 |
| INT4 w/o MW ECO + RTN | **49.15** | 71.59 | 16.30 | **78.88** | 81.34 | **38.63** |
| INT4 w/o MW ECO + SR | 48.55 | 71.17 | 16.00 | 78.84 | **81.50** | 38.41 |

### 4.6. Study on Quantization Schemes

To systematically study how ECO behaves across precisions and quantization granularities, we pre-train the 30M model on a Chinchilla-optimal number of tokens. We evaluate INT8, INT4, INT3, INT2, and INT1 weight-only quantization using QuEST's MSE-optimal scaling (Panferov et al., 2025), and consider three granularities: row-wise, group size 128, and group size 32. Table 3 reports the final validation loss.

Three observations stand out. First, consistent with our FP8 results, ECO recovers the master-weight baselines at INT8, making it the first method to achieve near-lossless 8-bit training without master weights. Second, as precision decreases from INT4 to INT1, ECO consistently and substantially outperforms naive master-weight removal, although the gap to the master-weight baseline widens as expected under increasingly severe quantization noise. Third, granularity (row-wise vs. group size 128 vs. group size 32) has little effect on the results, which we attribute to QuEST's MSE-optimal scaling combined with the near-Gaussian distribution of the weights.

### 4.7. Compatibility with Optimizer State Compression

A natural question is whether ECO remains effective when combined with optimizer state compression. We study the interaction between ECO and two popular first-moment

*Table 3.* Validation loss of the 30M model under weight-only integer quantization with QuEST's MSE-optimal scaling, across precisions (INT8–INT1) and granularities (row-wise, group size 128, group size 32). ECO matches the master-weight baselines at INT8 and consistently outperforms naive master-weight removal at lower precisions.

| | Row-wise | | | | |
| --- | --- | --- | --- | --- | --- |
| | INT8 | INT4 | INT3 | INT2 | INT1 |
| w/ MW + RTN | 3.4636 | 3.4944 | 3.5260 | 3.5985 | 3.9525 |
| w/ MW + SR | 3.4659 | 3.5183 | 3.5964 | 3.7323 | 4.0927 |
| w/o MW + SR | 3.5309 | 3.8255 | 3.9840 | 4.1158 | 5.1324 |
| w/o MW + SR + ECO | **3.4649** | **3.5445** | **3.6417** | **3.8273** | **4.7508** |

| | Group Size 128 | | | | |
| --- | --- | --- | --- | --- | --- |
| | INT8 | INT4 | INT3 | INT2 | INT1 |
| w/ MW + RTN | 3.4623 | 3.4897 | 3.5237 | 3.6084 | 3.9594 |
| w/ MW + SR | 3.4645 | 3.5193 | 3.5934 | 3.7289 | 4.1138 |
| w/o MW + SR | 3.5293 | 3.9420 | 4.2001 | 4.2232 | 5.1101 |
| w/o MW + SR + ECO | **3.4661** | **3.5514** | **3.6538** | **3.8673** | **4.7696** |

| | Group Size 32 | | | | |
| --- | --- | --- | --- | --- | --- |
| | INT8 | INT4 | INT3 | INT2 | INT1 |
| w/ MW + RTN | 3.4626 | 3.4892 | 3.5290 | 3.6009 | 3.9690 |
| w/ MW + SR | 3.4641 | 3.5126 | 3.5887 | 3.7317 | 4.1343 |
| w/o MW + SR | 3.5330 | 4.3067 | 4.6982 | 4.2519 | 5.1225 |
| w/o MW + SR + ECO | **3.4630** | **3.5684** | **3.6942** | **3.9855** | **4.7534** |

*Table 4.* Validation loss of the 30M model in FP8 row-wise format under different combinations of master-weight and first-moment quantization. The second Adam moment is kept in high precision throughout. ECO retains its near-lossless behavior when combined with 8-bit momentum compression.

| Validation Loss (30M) | FP32 Mom. | 8-bit (COAT) | 8-bit (8-bit Adam) |
| --- | --- | --- | --- |
| FP8 w/ MW + RTN | 3.4633 | 3.4632 | 3.4644 |
| FP8 w/ MW + SR | 3.4673 | 3.4663 | 3.4690 |
| FP8 w/o MW + SR | 3.5104 | 3.5075 | 3.5098 |
| FP8 w/o MW ECO + SR | **3.4705** | **3.4651** | **3.4684** |

*Table 5.* Validation loss of the 50M model in FP8 row-wise format under different combinations of master-weight and first-moment quantization. The second Adam moment is kept in high precision throughout. As with the 30M model, ECO remains near-lossless under 8-bit momentum compression.

| Validation Loss (50M) | FP32 Mom. | 8-bit (COAT) | 8-bit (8-bit Adam) |
| --- | --- | --- | --- |
| FP8 w/ MW + RTN | 3.2811 | 3.2809 | 3.2813 |
| FP8 w/ MW + SR | 3.2840 | 3.2831 | 3.2851 |
| FP8 w/o MW + SR | 3.3311 | 3.3343 | 3.3333 |
| FP8 w/o MW ECO + SR | **3.2842** | **3.2856** | **3.2827** |

## 5. Conclusion

ECO is the first general-purpose, scalable method for quantized LLM training without master weights. It removes high-precision accumulation by forming an error-feedback loop through the optimizer's momentum, with no additional memory overhead. Our analysis shows that ECO avoids the instability of naive master-weight removal. Empirically, across dense Transformers and SMoE models, ECO nearly matches high-precision baselines while improving the static-memory versus loss trade-off, showing that it can serve as a practical building block for future low-precision training.

**Limitations.** Both theory and experiments indicate that ECO performs best with stochastic rounding (SR). While SR is becoming more common in hardware, some devices only support round-to-nearest (RTN). In that setting, ECO still outperforms naive approaches but can exhibit a higher noise floor, consistent with our theory. Moreover, when master weights are available, RTN generally slightly outperforms SR in practice (Castro et al., 2025); in contrast, ECO relies on the unbiasedness of SR for its strongest guarantees. This introduces a slight accuracy ceiling relative to the best RTN-based master-weight baselines.

## Impact Statement

This paper presents work whose goal is to advance the field of Machine Learning. There are many potential societal consequences of our work, none which we feel must be specifically highlighted here.

compression methods, COAT (Xi et al., 2024a) and 8-bit Adam (Dettmers et al., 2021). We pre-train the 30M and 50M models in FP8 row-wise format on a Chinchilla-optimal number of tokens. For this study we keep the second Adam moment in high precision, as it has no direct interaction with ECO, and vary the precision of the master weights and the first moment. Tables 4 and 5 report the validation loss for the 30M and 50M models, respectively.

**Compatibility.** The results show that ECO and first-moment compression are compatible in the 8-bit setting: applying ECO on top of 8-bit momentum (via either COAT or 8-bit Adam) preserves the near-lossless behavior observed with FP32 momentum, recovering the master-weight baselines in both model sizes. ECO can therefore be stacked on existing optimizer state compression to obtain additional memory savings.

**Memory.** The savings compound across the weight and optimizer buffers. A standard master-weight setup with an FP32 optimizer stores 4 bytes/param for weights, 4 for the first moment, and 4 for the second moment, totaling 12 bytes/param. ECO with an FP32 optimizer removes the master weights (1 byte/param weight), reducing this to 9 bytes/param, a 25% reduction. Adding 8-bit first-moment compression on top of ECO further drops the first moment to 1 byte/param, reaching 6 bytes/param, a 50% reduction. Although we leave it to future work, additionally compressing the second moment to 8-bit would yield a 75% reduction relative to the baseline.

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

# A. Exact Error Injection

This appendix shows that SGDM with high-precision master weights can be reproduced *exactly* using only quantized weights, provided the momentum buffer receives an "ideal" correction that depends on both the current and previous quantization residuals.

**SGDM with Master Weights.** Let $q(\cdot)$ be the weight quantizer. Let $\boldsymbol{\theta}_t$ denote the high-precision master weights, and let $\hat{\boldsymbol{\theta}}_t^{\mathrm{MW}} \leftarrow q(\boldsymbol{\theta}_t)$ be the quantized weights used for the forward/backward pass at step $t$. Using the gradient

$$\mathbf{g}_t \leftarrow \nabla f(\hat{\boldsymbol{\theta}}_t^{\mathrm{MW}}), \tag{21}$$

SGDM with master weights updates

$$\mathbf{m}_{t+1}^{\mathrm{MW}} \leftarrow \beta \, \mathbf{m}_t^{\mathrm{MW}} + (1 - \beta) \, \mathbf{g}_t, \tag{22}$$

$$\boldsymbol{\theta}_{t+1} \leftarrow \boldsymbol{\theta}_t - \eta \, \mathbf{m}_{t+1}^{\mathrm{MW}}, \tag{23}$$

$$\hat{\boldsymbol{\theta}}_{t+1}^{\mathrm{MW}} \leftarrow q(\boldsymbol{\theta}_{t+1}), \tag{24}$$

and we define the quantization residual of $\boldsymbol{\theta}_{t+1}$ as

$$\mathbf{e}_{t+1}^{\mathrm{MW}} := \boldsymbol{\theta}_{t+1} - \hat{\boldsymbol{\theta}}_{t+1}^{\mathrm{MW}}. \tag{25}$$

**No-Master-Weight SGDM with Ideal Momentum Injection.** This variant stores only quantized weights $\hat{\boldsymbol{\theta}}_t^{\mathrm{IM}}$, a momentum buffer $\mathbf{m}_t^{\mathrm{IM}}$, and the previous residual $\mathbf{e}_t^{\mathrm{IM}}$. At step $t$, it computes

$$\mathbf{g}_t \leftarrow \nabla f(\hat{\boldsymbol{\theta}}_t^{\mathrm{IM}}),$$

$$\bar{\mathbf{m}}_{t+1} \leftarrow \beta \, \mathbf{m}_t^{\mathrm{IM}} + (1 - \beta) \, \mathbf{g}_t, \tag{26}$$

$$\tilde{\boldsymbol{\theta}}_{t+1} \leftarrow \hat{\boldsymbol{\theta}}_t^{\mathrm{IM}} - \eta \, \bar{\mathbf{m}}_{t+1}, \tag{27}$$

$$\hat{\boldsymbol{\theta}}_{t+1}^{\mathrm{IM}} \leftarrow q(\tilde{\boldsymbol{\theta}}_{t+1}), \tag{28}$$

$$\mathbf{e}_{t+1}^{\mathrm{IM}} \leftarrow \tilde{\boldsymbol{\theta}}_{t+1} - \hat{\boldsymbol{\theta}}_{t+1}^{\mathrm{IM}}, \tag{29}$$

$$\mathbf{m}_{t+1}^{\mathrm{IM}} \leftarrow \bar{\mathbf{m}}_{t+1} + \frac{1}{\eta} \mathbf{e}_t^{\mathrm{IM}} - \frac{1}{\eta\beta} \mathbf{e}_{t+1}^{\mathrm{IM}}. \tag{30}$$

**Theorem (Exact equivalence).** Assume SGDM with master weights starts from $(\boldsymbol{\theta}_0, \mathbf{m}_0^{\mathrm{MW}})$. Initialize the injected method by

$$\hat{\boldsymbol{\theta}}_0^{\mathrm{IM}} \leftarrow q(\boldsymbol{\theta}_0), \qquad \mathbf{e}_0^{\mathrm{IM}} \leftarrow \boldsymbol{\theta}_0 - \hat{\boldsymbol{\theta}}_0^{\mathrm{IM}}, \qquad \mathbf{m}_0^{\mathrm{IM}} \leftarrow \mathbf{m}_0^{\mathrm{MW}} - \frac{1}{\eta\beta} \mathbf{e}_0^{\mathrm{IM}}. \tag{31}$$

Then, for all $t \geq 0$, the quantized iterates produced by the injected method satisfy

$$\hat{\boldsymbol{\theta}}_t^{\mathrm{IM}} = \hat{\boldsymbol{\theta}}_t^{\mathrm{MW}},$$

and therefore the two procedures produce identical gradients at every step.

**Proof.** Define the *implicit* master weights and momentum corresponding to the injected method by

$$\boldsymbol{\theta}_t^{\star} := \hat{\boldsymbol{\theta}}_t^{\mathrm{IM}} + \mathbf{e}_t^{\mathrm{IM}}, \qquad \mathbf{m}_t^{\star} := \mathbf{m}_t^{\mathrm{IM}} + \frac{1}{\eta\beta} \mathbf{e}_t^{\mathrm{IM}}. \tag{32}$$

By (31), we have $\boldsymbol{\theta}_0^{\star} = \boldsymbol{\theta}_0$ and $\mathbf{m}_0^{\star} = \mathbf{m}_0^{\mathrm{MW}}$.

From (29), we have

$$\tilde{\boldsymbol{\theta}}_{t+1} = \hat{\boldsymbol{\theta}}_{t+1}^{\mathrm{IM}} + \mathbf{e}_{t+1}^{\mathrm{IM}}. \tag{33}$$

Hence,

$$\boldsymbol{\theta}_{t+1}^{\star} = \hat{\boldsymbol{\theta}}_{t+1}^{\text{IM}} + \mathbf{e}_{t+1}^{\text{IM}} = \tilde{\boldsymbol{\theta}}_{t+1}. \tag{34}$$

Combining with (28), we get $\hat{\boldsymbol{\theta}}_{t+1}^{\text{IM}} \leftarrow q(\boldsymbol{\theta}_{t+1}^{\star})$.

Next, using (30) and (32),

$$\begin{aligned}
\mathbf{m}_{t+1}^{\star} &= \mathbf{m}_{t+1}^{\text{IM}} + \frac{1}{\eta\beta}\mathbf{e}_{t+1}^{\text{IM}} \\
&= \bar{\mathbf{m}}_{t+1} + \frac{1}{\eta}\mathbf{e}_t^{\text{IM}} - \frac{1}{\eta\beta}\mathbf{e}_{t+1}^{\text{IM}} + \frac{1}{\eta\beta}\mathbf{e}_{t+1}^{\text{IM}} \\
&= \bar{\mathbf{m}}_{t+1} + \frac{1}{\eta}\mathbf{e}_t^{\text{IM}} \\
&= \beta\,\mathbf{m}_t^{\text{IM}} + (1-\beta)\,\mathbf{g}_t + \frac{1}{\eta}\mathbf{e}_t^{\text{IM}} \tag{35} \\
&= \beta\Big(\mathbf{m}_t^{\text{IM}} + \frac{1}{\eta\beta}\mathbf{e}_t^{\text{IM}}\Big) + (1-\beta)\,\mathbf{g}_t \\
&= \beta\,\mathbf{m}_t^{\star} + (1-\beta)\,\mathbf{g}_t. \tag{36}
\end{aligned}$$

Thus $\mathbf{m}_{t+1}^{\star}$ follows the same SGDM momentum recurrence as (22).

Finally, using (34), (27), and (35),

$$\begin{aligned}
\boldsymbol{\theta}_{t+1}^{\star} = \tilde{\boldsymbol{\theta}}_{t+1} = \hat{\boldsymbol{\theta}}_t^{\text{IM}} - \eta\,\bar{\mathbf{m}}_{t+1} \\
= (\hat{\boldsymbol{\theta}}_t^{\text{IM}} + \mathbf{e}_t^{\text{IM}}) - \eta\big(\bar{\mathbf{m}}_{t+1} + \tfrac{1}{\eta}\mathbf{e}_t^{\text{IM}}\big) \\
= \boldsymbol{\theta}_t^{\star} - \eta\,\mathbf{m}_{t+1}^{\star}, \tag{37}
\end{aligned}$$

which matches the master-weight update (23). Therefore, with identical initial conditions, the implicit variables $(\boldsymbol{\theta}_t^{\star}, \mathbf{m}_t^{\star})$ evolve exactly as SGDM with master weights, implying

$$\hat{\boldsymbol{\theta}}_t^{\text{IM}} = q(\boldsymbol{\theta}_t^{\star}) = q(\boldsymbol{\theta}_t) = \hat{\boldsymbol{\theta}}_t^{\text{MW}} \qquad \text{for all } t.$$

$\square$

## B. Convergence Proofs

### B.1. Proof of Lemma 3.5

*Proof.* First, substitute $\tilde{\mathbf{m}}_{t+1}$ from Eq. (6) into Eq. (3):

$$\tilde{\boldsymbol{\theta}}_{t+1} = \hat{\boldsymbol{\theta}}_t - \eta(\hat{\mathbf{m}}_{t+1} - \alpha\mathbf{e}_{t+1}). \tag{38}$$

Using $\mathbf{e}_{t+1} = \tilde{\boldsymbol{\theta}}_{t+1} - \hat{\boldsymbol{\theta}}_{t+1}$, we rearrange to solve for $\hat{\boldsymbol{\theta}}_{t+1}$:

$$\begin{aligned}
\hat{\boldsymbol{\theta}}_{t+1} + \mathbf{e}_{t+1} &= \hat{\boldsymbol{\theta}}_t - \eta\hat{\mathbf{m}}_{t+1} + \eta\alpha\mathbf{e}_{t+1} \\
\hat{\boldsymbol{\theta}}_{t+1} &= \hat{\boldsymbol{\theta}}_t - \eta\hat{\mathbf{m}}_{t+1} - (1-\eta\alpha)\mathbf{e}_{t+1}. \tag{39}
\end{aligned}$$

Substituting $\alpha = \frac{1}{\eta}(1 - \frac{1}{\beta})$, we have $1 - \eta\alpha = \frac{1}{\beta}$. Thus:

$$\hat{\boldsymbol{\theta}}_{t+1} = \hat{\boldsymbol{\theta}}_t - \eta\hat{\mathbf{m}}_{t+1} - \frac{1}{\beta}\mathbf{e}_{t+1}. \tag{40}$$

Now, examine the update of the virtual sequence $\boldsymbol{\theta}_{t+1}$:

$$\boldsymbol{\theta}_{t+1} = \hat{\boldsymbol{\theta}}_{t+1} - \frac{\eta\beta}{1-\beta}\hat{\mathbf{m}}_{t+1}. \tag{41}$$

We expand this expression:

$$\boldsymbol{\theta}_{t+1} = \hat{\boldsymbol{\theta}}_{t+1} - \frac{\eta\beta}{1-\beta}\hat{\mathbf{m}}_{t+1}$$

$$= (\tilde{\boldsymbol{\theta}}_{t+1} - \mathbf{e}_{t+1}) - \frac{\eta\beta}{1-\beta}(\tilde{\mathbf{m}}_{t+1} + \alpha\mathbf{e}_{t+1})$$

$$= (\hat{\boldsymbol{\theta}}_t - \eta\tilde{\mathbf{m}}_{t+1} - \mathbf{e}_{t+1}) - \frac{\eta\beta}{1-\beta}\tilde{\mathbf{m}}_{t+1} - \frac{\eta\beta\alpha}{1-\beta}\mathbf{e}_{t+1}$$

$$= \hat{\boldsymbol{\theta}}_t - \eta\left(1 + \frac{\beta}{1-\beta}\right)\tilde{\mathbf{m}}_{t+1} - \left(1 + \frac{\eta\beta\alpha}{1-\beta}\right)\mathbf{e}_{t+1}. \tag{42}$$

Using the identity $1 + \frac{\beta}{1-\beta} = \frac{1}{1-\beta}$, the coefficient of $\tilde{\mathbf{m}}_{t+1}$ is $-\frac{\eta}{1-\beta}$. Now check the coefficient of $\mathbf{e}_{t+1}$. Using $\alpha = \frac{\beta-1}{\eta\beta} = -\frac{1-\beta}{\eta\beta}$:

$$1 + \frac{\eta\beta}{1-\beta}\left(-\frac{1-\beta}{\eta\beta}\right) = 1 - 1 = 0. \tag{43}$$

The error term $\mathbf{e}_{t+1}$ vanishes perfectly. We are left with:

$$\boldsymbol{\theta}_{t+1} = \hat{\boldsymbol{\theta}}_t - \frac{\eta}{1-\beta}\tilde{\mathbf{m}}_{t+1}. \tag{44}$$

Expanding $\tilde{\mathbf{m}}_{t+1} = \beta\hat{\mathbf{m}}_t + (1-\beta)\nabla f(\hat{\boldsymbol{\theta}}_t)$:

$$\boldsymbol{\theta}_{t+1} = \hat{\boldsymbol{\theta}}_t - \frac{\eta\beta}{1-\beta}\hat{\mathbf{m}}_t - \eta\nabla f(\hat{\boldsymbol{\theta}}_t)$$

$$= \boldsymbol{\theta}_t - \eta\nabla f(\hat{\boldsymbol{\theta}}_t). \tag{45}$$

$\square$

## B.2. Proof of Lemma 3.6

*Proof.* Applying the standard descent lemma to the virtual sequence $\boldsymbol{\theta}_t$:

$$f(\boldsymbol{\theta}_{t+1}) \leq f(\boldsymbol{\theta}_t) + \langle\nabla f(\boldsymbol{\theta}_t), \boldsymbol{\theta}_{t+1} - \boldsymbol{\theta}_t\rangle + \frac{L}{2}\|\boldsymbol{\theta}_{t+1} - \boldsymbol{\theta}_t\|_2^2$$

$$\leq f(\boldsymbol{\theta}_t) - \eta\langle\nabla f(\boldsymbol{\theta}_t), \nabla f(\hat{\boldsymbol{\theta}}_t)\rangle + \frac{L\eta^2}{2}\left\|\nabla f(\hat{\boldsymbol{\theta}}_t)\right\|_2^2. \tag{46}$$

Using the identity $-\langle a, b\rangle = -\frac{1}{2}\|a\|_2^2 - \frac{1}{2}\|b\|_2^2 + \frac{1}{2}\|a-b\|_2^2$:

$$f(\boldsymbol{\theta}_{t+1}) \leq f(\boldsymbol{\theta}_t) - \frac{\eta}{2}\|\nabla f(\boldsymbol{\theta}_t)\|_2^2 - \frac{\eta}{2}(1 - L\eta)\left\|\nabla f(\hat{\boldsymbol{\theta}}_t)\right\|_2^2 + \frac{\eta}{2}\left\|\nabla f(\boldsymbol{\theta}_t) - \nabla f(\hat{\boldsymbol{\theta}}_t)\right\|_2^2. \tag{47}$$

The term with $\|\nabla f(\boldsymbol{\theta}_t)\|_2^2$ is non-positive. Additionally, as $\eta \leq \frac{1}{2L}$, we have $1 - L\eta \geq \frac{1}{2}$. Using $L$-smoothness on the last term:

$$f(\boldsymbol{\theta}_{t+1}) \leq f(\boldsymbol{\theta}_t) - \frac{\eta}{4}\left\|\nabla f(\hat{\boldsymbol{\theta}}_t)\right\|_2^2 + \frac{\eta L^2}{2}\left\|\boldsymbol{\theta}_t - \hat{\boldsymbol{\theta}}_t\right\|_2^2. \tag{48}$$

The difference between the virtual and actual (quantized) parameters is:

$$\boldsymbol{\theta}_t - \hat{\boldsymbol{\theta}}_t = -\frac{\eta\beta}{1-\beta}\hat{\mathbf{m}}_t. \tag{49}$$

Substituting $C = \frac{\eta\beta}{1-\beta}$ yields $\left\|\boldsymbol{\theta}_t - \hat{\boldsymbol{\theta}}_t\right\|_2^2 = C^2\|\hat{\mathbf{m}}_t\|_2^2$. Substituting into the inequality gives:

$$f(\boldsymbol{\theta}_{t+1}) \leq f(\boldsymbol{\theta}_t) - \frac{\eta}{4}\left\|\nabla f(\hat{\boldsymbol{\theta}}_t)\right\|_2^2 + \frac{\eta L^2 C^2}{2}\|\hat{\mathbf{m}}_t\|_2^2. \tag{50}$$

$\square$

## B.3. Proof of Lemma 3.7

*Proof.* We expand the recursion for $\hat{\mathbf{m}}_t$ starting from $\hat{\mathbf{m}}_0 = 0$. With the updated update rule $\tilde{\mathbf{m}}_{t+1} = \beta\hat{\mathbf{m}}_t + (1-\beta)\nabla f(\hat{\boldsymbol{\theta}}_t)$, the expansion becomes:

$$\hat{\mathbf{m}}_t = \sum_{k=1}^{t} \beta^{t-k}\left((1-\beta)\nabla f(\hat{\boldsymbol{\theta}}_{k-1}) + \alpha\mathbf{e}_k\right). \tag{51}$$

We define two components, the gradient accumulation $S_1$ and the error accumulation $S_2$:

$$S_1 = (1-\beta)\sum_{k=1}^{t}\beta^{t-k}\nabla f(\hat{\boldsymbol{\theta}}_{k-1}), \quad S_2 = \sum_{k=1}^{t}\beta^{t-k}\alpha\mathbf{e}_k. \tag{52}$$

Using the inequality $\|a+b\|^2 \le 2\|a\|^2 + 2\|b\|^2$, we have $\mathbb{E}[\|\hat{\mathbf{m}}_t\|^2] \le 2\mathbb{E}[\|S_1\|^2] + 2\mathbb{E}[\|S_2\|^2]$.

For the gradient term $S_1$, we use the deterministic triangle inequality bound. The $(1-\beta)$ factor scales the sum:

$$\|S_1\| \le (1-\beta)\sum_{k=1}^{t}\beta^{t-k}\|\nabla f(\hat{\boldsymbol{\theta}}_{k-1})\| \le G(1-\beta)\sum_{j=0}^{t-1}\beta^j. \tag{53}$$

Using the geometric series sum bound $\sum_{j=0}^{t-1}\beta^j \le \frac{1}{1-\beta}$, the terms cancel nicely:

$$\|S_1\| \le G(1-\beta)\frac{1}{1-\beta} = G. \tag{54}$$

Thus $\mathbb{E}[\|S_1\|^2] \le G^2$.

For the error term $S_2$, utilizing the unbiasedness assumption where $\mathbb{E}[\mathbf{e}_k|\mathbf{e}_j] = 0$ for $k > j$:

$$\mathbb{E}[\|S_2\|^2] = \mathbb{E}\left[\left\|\sum_{k=1}^{t}\beta^{t-k}\alpha\mathbf{e}_k\right\|^2\right]$$

$$= \sum_{k=1}^{t}\beta^{2(t-k)}\alpha^2\mathbb{E}[\|\mathbf{e}_k\|^2] + \sum_{j\neq k}\text{Cross Terms}$$

$$= \sum_{k=1}^{t}\beta^{2(t-k)}\alpha^2\mathbb{E}[\|\mathbf{e}_k\|^2]. \tag{55}$$

Using $\mathbb{E}[\|\mathbf{e}_k\|^2] \le \sigma^2$, we bound the sum by the infinite geometric series with ratio $\beta^2$:

$$\mathbb{E}[\|S_2\|^2] \le \alpha^2\sigma^2\sum_{j=0}^{\infty}(\beta^2)^j = \frac{\alpha^2\sigma^2}{1-\beta^2}. \tag{56}$$

Combining these results:

$$\mathbb{E}[\|\hat{\mathbf{m}}_t\|^2] \le 2G^2 + \frac{2\alpha^2\sigma^2}{1-\beta^2}. \tag{57}$$

$\square$

## B.4. Proof of Theorem 3.8

*Proof.* We take expectations from both sides of the descent Lemma 3.6 and substitute the momentum bound $M^2$ (Lemma 3.7).

$$\mathbb{E}\left[f(\boldsymbol{\theta}_{t+1})\right] \le \mathbb{E}\left[f(\boldsymbol{\theta}_t)\right] - \frac{\eta}{4}\mathbb{E}\left[\|\nabla f(\hat{\boldsymbol{\theta}}_t)\|^2\right] + \frac{\eta L^2 C^2}{2}M^2. \tag{58}$$

Rearranging to isolate the gradient norm:

$$\frac{\eta}{4}\mathbb{E}\left[\|\nabla f(\hat{\boldsymbol{\theta}}_t)\|^2\right] \le \mathbb{E}\left[f(\boldsymbol{\theta}_t) - f(\boldsymbol{\theta}_{t+1})\right] + \frac{\eta L^2 C^2}{2}M^2. \tag{59}$$

Summing from $t = 0$ to $T - 1$:

$$\frac{\eta}{4} \sum_{t=0}^{T-1} \mathbb{E}\left[\|\nabla f(\hat{\boldsymbol{\theta}}_t)\|^2\right] \leq \mathbb{E}\left[f(\boldsymbol{\theta}_0) - f(\boldsymbol{\theta}_T)\right] + \sum_{t=0}^{T-1} \frac{\eta L^2 C^2}{2} M^2$$

$$\leq f(\boldsymbol{\theta}_0) - f^* + T\frac{\eta L^2 C^2}{2} M^2. \tag{60}$$

Dividing by $T\eta/4$:

$$\frac{1}{T} \sum_{t=0}^{T-1} \mathbb{E}\left[\|\nabla f(\hat{\boldsymbol{\theta}}_t)\|^2\right] \leq \frac{4(f(\boldsymbol{\theta}_0) - f^*)}{\eta T} + 2L^2 C^2 M^2. \tag{61}$$

Defining $\sigma_{\text{quant}}^2 = 2L^2 C^2 M^2$ yields the final result. $\qquad \square$

### B.5. Proof of Lemma 3.9

*Proof.* Let $\|\mathbf{e}_t\| \leq \delta$ (absolute error bound).

$$\|\hat{\mathbf{m}}_{t+1}\| \leq \beta\|\hat{\mathbf{m}}_t\| + (1 - \beta)\|\nabla f(\hat{\boldsymbol{\theta}}_t)\| + |\alpha|\|\mathbf{e}_{t+1}\|. \tag{62}$$

Using the bounded gradient assumption $\|\nabla f(\boldsymbol{\theta})\| \leq G$:

$$\|\hat{\mathbf{m}}_{t+1}\| \leq \beta\|\hat{\mathbf{m}}_t\| + (1 - \beta)G + |\alpha|\delta. \tag{63}$$

This is a linear recurrence of the form $x_{t+1} \leq \beta x_t + K$. Assuming $\hat{\mathbf{m}}_0 = 0$, the sequence is bounded by the sum of the geometric series:

$$\|\hat{\mathbf{m}}_t\| \leq \sum_{i=0}^{t} \beta^i((1-\beta)G + |\alpha|\delta) \leq G + \frac{|\alpha|\delta}{1-\beta} := M. \tag{64}$$

$$\square$$

### B.6. Proof of Theorem 3.10

*Proof.* We use Lemmas 3.6 and 3.9.

Summing the descent inequality from $t = 0$ to $T - 1$:

$$f(\boldsymbol{\theta}_T) \leq f(\boldsymbol{\theta}_0) - \frac{\eta}{4} \sum_{t=0}^{T-1} \left\|\nabla f(\hat{\boldsymbol{\theta}}_t)\right\|_2^2 + \frac{\eta T L^2 C^2}{2} M_{\text{det}}^2. \tag{65}$$

Rearranging and using $f^* \leq f(\boldsymbol{\theta}_T)$:

$$\frac{1}{T} \sum_{t=0}^{T-1} \left\|\nabla f(\hat{\boldsymbol{\theta}}_t)\right\|_2^2 \leq \frac{4(f(\boldsymbol{\theta}_0) - f^*)}{\eta T} + 2L^2 C^2 M_{\text{det}}^2. \tag{66}$$

Defining $\Gamma_{\text{quant}}^2 = 2L^2 C^2 M_{\text{det}}^2$ completes the proof. $\qquad \square$

## C. Formal Analysis of the Worst-Case Lower-Bounds

This appendix provides proofs for the claims made in Section 3.4.

All three regimes considered below can be written in the linear form

$$\begin{aligned} x_{t+1} &= ax_t + bm_t + B_1\xi_{t+1}, \\ m_{t+1} &= cx_t + dm_t + B_2\xi_{t+1}, \end{aligned} \tag{67}$$

for constants $a, b, c, d, B_1, B_2$ that depend on the regime. Define the second moments

$$u_t = \mathbb{E}[x_t^2], \qquad v_t = \mathbb{E}[x_t m_t], \qquad w_t = \mathbb{E}[m_t^2]. \tag{68}$$

**Lemma C.1** (Second-moment update equations). *The dynamics* (67) *imply*

$$u_{t+1} = a^2 u_t + 2ab\, v_t + b^2 w_t + B_1^2 \sigma^2, \tag{69}$$

$$v_{t+1} = ac\, u_t + (ad + bc)\, v_t + bd\, w_t + B_1 B_2 \sigma^2, \tag{70}$$

$$w_{t+1} = c^2 u_t + 2cd\, v_t + d^2 w_t + B_2^2 \sigma^2. \tag{71}$$

*Proof.* Expand each square/product and remove all cross terms. For example, for $u_{t+1}$:

$$u_{t+1} = \mathbb{E}[(ax_t + bm_t + B_1 \xi_{t+1})^2] = a^2 u_t + 2abv_t + b^2 w_t + B_1^2 \mathbb{E}[\xi_{t+1}^2],$$

since $\mathbb{E}[x_t \xi_{t+1}] = \mathbb{E}[m_t \xi_{t+1}] = 0$ and $\mathbb{E}[\xi_{t+1}^2] = \sigma^2$. The proofs for $v_{t+1}$ and $w_{t+1}$ are identical. $\square$

**Stability.** Let $A = \begin{pmatrix} a & b \\ c & d \end{pmatrix}$ denote the deterministic part of (67). A sufficient and standard condition for existence of a unique stationary second moment is $\rho(A) < 1$, where $\rho(A)$ indicates $A$'s largest absolute eigenvalue. For the SGDM parameters used below, this holds whenever

$$0 < \eta < \frac{2(1 + \beta)}{(1 - \beta)L}. \tag{72}$$

All stationary calculations below assume (72), which also guarantees that the denominators appearing in the closed forms are strictly positive.

## C.1. Fundamental limits on $f(x) = \frac{L}{2} x^2$

We now analyze the stationary squared gradient of the *quantized* parameter used by the model. For any regime, define the (steady-state) metric

$$\mathcal{L} := \lim_{t \to \infty} \mathbb{E}[g(\hat{x}_t)^2] = L^2 \lim_{t \to \infty} \mathbb{E}[\hat{x}_t^2], \tag{73}$$

where $\hat{x}_t$ is the parameter seen by the forward/backward pass (quantized weights).

### C.1.1. SGDM WITH MASTER WEIGHTS

**Algorithm.** We store a full-precision master weight $x_t$. Each step quantizes it for the gradient:

$$\hat{x}_t = q(x_t) = x_t + \xi_t, \tag{74}$$

then performs SGDM using $\hat{x}_t$:

$$m_{t+1} = \beta m_t + (1 - \beta)L\hat{x}_t, \qquad x_{t+1} = x_t - \eta m_{t+1}. \tag{75}$$

**Linear form.** Let $c := (1 - \beta)L$, and define

$$a := 1 - \eta c, \qquad b := -\eta \beta, \qquad d := \beta. \tag{76}$$

Using $\hat{x}_t = x_t + \xi_t$, we obtain

$$\begin{aligned} x_{t+1} &= ax_t + bm_t + (-\eta c)\, \xi_t, \\ m_{t+1} &= cx_t + dm_t + c\, \xi_t. \end{aligned} \tag{77}$$

This matches (67) with $(B_1, B_2) = (-\eta c,\ c)$.

**Stationary second moments.** Let $(u, v, w)$ denote the stationary solution of (69)–(71). Plugging $B_1 = -\eta c$ and $B_2 = c$ into Lemma C.1 and setting $(u_{t+1}, v_{t+1}, w_{t+1}) = (u, v, w)$ yields the linear system

$$u = a^2 u + 2abv + b^2 w + \eta^2 c^2 \sigma^2, \tag{78}$$

$$v = ac\, u + (ad + bc)\, v + bd\, w - \eta c^2 \sigma^2, \tag{79}$$

$$w = c^2 u + 2cd\, v + d^2 w + c^2 \sigma^2. \tag{80}$$

We solve it by elimination.

From (80) and $d = \beta$,

$$(1 - \beta^2)w = c^2(u + \sigma^2) + 2c\beta v \quad \Longrightarrow \quad w = \frac{c^2(u + \sigma^2) + 2c\beta v}{1 - \beta^2}. \tag{81}$$

Substitute (81) into (79). Using $ad + bc = \beta(1 - \eta c) + (-\eta\beta)c = \beta - 2\eta\beta c$ and $bd = b\beta = -\eta\beta^2$, we rewrite (79) as

$$v = ac\,u + (\beta - 2\eta\beta c)\,v - \eta\beta^2 w - \eta c^2 \sigma^2. \tag{82}$$

Move the $v$ and $w$ terms to the left and substitute $w$ from (81). This yields a single linear equation in $v$ and $u$, which solves to

$$v = \frac{L^2 \eta \sigma^2 (\beta - 1)}{2(1 + \beta) - L\eta(1 - \beta)}. \tag{83}$$

Plugging (83) back into (81) gives

$$w = \frac{2L^2 \sigma^2 (1 - \beta)}{2(1 + \beta) - L\eta(1 - \beta)}. \tag{84}$$

Finally, substitute (83) and (84) into (78). Solving for $u$ yields

$$u = \mathbb{E}[x^2] = \frac{L\eta \sigma^2 (1 + \beta)}{2(1 + \beta) - L\eta(1 - \beta)}. \tag{85}$$

**Limit of the squared gradient.** The model uses $\hat{x} = x + \xi$ with $\mathbb{E}[x\xi] = 0$. Hence

$$\mathbb{E}[\hat{x}^2] = \mathbb{E}[x^2] + \mathbb{E}[\xi^2] = u + \sigma^2. \tag{86}$$

Therefore the stationary squared gradient satisfies

$$\mathcal{L}_{\mathrm{MW}} = L^2(u + \sigma^2). \tag{87}$$

Taking $\eta \to 0$ in (85) gives $u \to 0$, so

$$\lim_{\eta \to 0} \mathcal{L}_{\mathrm{MW}} = L^2 \sigma^2. \tag{88}$$

### C.1.2. NAIVE MASTER-WEIGHT REMOVAL

**Algorithm.** We store only quantized weights $\hat{x}_t$. Each step:

$$m_{t+1} = \beta m_t + (1 - \beta)L\hat{x}_t, \qquad \tilde{x}_{t+1} = \hat{x}_t - \eta m_{t+1}, \qquad \hat{x}_{t+1} = q(\tilde{x}_{t+1}) = \tilde{x}_{t+1} + \xi_{t+1}. \tag{89}$$

**Linear form.** With $c = (1 - \beta)L$ and the same $a, b, d$ as above,

$$\begin{aligned} \hat{x}_{t+1} &= a\hat{x}_t + bm_t + 1 \cdot \xi_{t+1}, \\ m_{t+1} &= c\hat{x}_t + dm_t. \end{aligned} \tag{90}$$

This matches (67) with $(B_1, B_2) = (1, 0)$ and state $x_t \equiv \hat{x}_t$.

**Stationary second moments.** Let $(u, v, w)$ denote the stationary solution for $u = \mathbb{E}[\hat{x}^2]$. Plugging $(B_1, B_2) = (1, 0)$ into Lemma C.1 and setting stationarity yields

$$u = a^2 u + 2abv + b^2 w + \sigma^2, \tag{91}$$
$$v = ac\,u + (ad + bc)\,v + bd\,w, \tag{92}$$
$$w = c^2 u + 2cd\,v + d^2 w. \tag{93}$$

From (93) and $d = \beta$,

$$(1 - \beta^2)w = c^2 u + 2c\beta v \quad \Longrightarrow \quad w = \frac{c^2 u + 2c\beta v}{1 - \beta^2}. \tag{94}$$

Substitute (94) into (92); as above, $ad + bc = \beta - 2\eta\beta c$ and $bd = -\eta\beta^2$. This yields one linear equation in $(u, v)$, which solves to

$$v = -\frac{\sigma^2(L\eta - \beta - 1)}{\eta\big(2(1 + \beta) - L\eta(1 - \beta)\big)}. \tag{95}$$

Plugging (95) into (94) gives $w$; substituting $(v, w)$ into (91) and solving for $u$ yields the closed form

$$u = \mathbb{E}[\hat{x}^2] = \sigma^2 \frac{(1 - \beta^2) + 2\beta L\eta}{L\eta\big(2(1 - \beta^2) - L\eta(1 - \beta)^2\big)}. \tag{96}$$

**Divergence as $\eta \to 0$.** From (96), as $\eta \to 0$ the denominator is $2L\eta(1 - \beta^2) + o(\eta)$ while the numerator is $(1 - \beta^2) + o(1)$, hence

$$\mathbb{E}[\hat{x}^2] = \frac{\sigma^2}{2L\eta} + O(1), \qquad \eta \to 0. \tag{97}$$

Therefore

$$\mathcal{L}_{\text{Naive}} = L^2 \mathbb{E}[\hat{x}^2] \sim \frac{L\sigma^2}{2\eta} \xrightarrow[\eta \to 0]{} \infty. \tag{98}$$

### C.1.3. ECO: MOMENTUM INJECTION ELIMINATES THE $1/\eta$ BLOW-UP

**Algorithm.** ECO uses the same SGDM step as the naive method to compute $(\tilde{x}_{t+1}, \tilde{m}_{t+1})$ from $(\hat{x}_t, \hat{m}_t)$, then quantizes and injects the quantization error into momentum. Concretely:

$$\tilde{m}_{t+1} = \beta\hat{m}_t + (1 - \beta)L\hat{x}_t, \qquad \tilde{x}_{t+1} = \hat{x}_t - \eta\tilde{m}_{t+1}, \qquad \hat{x}_{t+1} = q(\tilde{x}_{t+1}) = \tilde{x}_{t+1} + \xi_{t+1}. \tag{99}$$

Define the (post-quantization) error $e_{t+1} := \tilde{x}_{t+1} - \hat{x}_{t+1} = -\xi_{t+1}$. ECO then sets

$$\hat{m}_{t+1} = \tilde{m}_{t+1} + \alpha e_{t+1} = \tilde{m}_{t+1} - \alpha\xi_{t+1}, \qquad \alpha = \frac{1}{\eta}\left(1 - \frac{1}{\beta}\right) = \frac{\beta - 1}{\eta\beta}. \tag{100}$$

Since $\beta \in (0, 1)$, $\alpha < 0$. Define the positive injection gain

$$\gamma := -\alpha = \frac{1 - \beta}{\eta\beta} > 0, \tag{101}$$

so that $\hat{m}_{t+1} = \tilde{m}_{t+1} + \gamma\xi_{t+1}$.

**Linear form.** With $c = (1 - \beta)L$ and the same $a, b, d$ as above, ECO becomes

$$\begin{aligned} \hat{x}_{t+1} &= a\hat{x}_t + b\hat{m}_t + 1 \cdot \xi_{t+1}, \\ \hat{m}_{t+1} &= c\hat{x}_t + d\hat{m}_t + \gamma\,\xi_{t+1}, \end{aligned} \tag{102}$$

i.e., (67) with $(B_1, B_2) = (1, \gamma)$.

**Stationary second moments.** Applying Lemma C.1 to (102) and setting stationarity yields

$$u = a^2 u + 2abv + b^2 w + \sigma^2, \tag{103}$$
$$v = ac\,u + (ad + bc)\,v + bd\,w + \gamma\sigma^2, \tag{104}$$
$$w = c^2 u + 2cd\,v + d^2 w + \gamma^2\sigma^2. \tag{105}$$

We again eliminate $w$ using (105) (same algebra as before) and then eliminate $v$ using (104). The resulting expressions simplify dramatically because $\gamma$ is coupled to $(\eta, \beta)$ by (101). Solving (103)–(105) yields the closed form

$$u = \mathbb{E}[\hat{x}^2] = \frac{2\sigma^2}{2(1 - \beta^2) - L\eta(1 - \beta)^2}. \tag{106}$$

**Finite noise floor as $\eta \to 0$.** Taking $\eta \to 0$ in (106) gives

$$\lim_{\eta \to 0} \mathbb{E}[\hat{x}^2] = \frac{\sigma^2}{1 - \beta^2}, \tag{107}$$

and therefore the stationary squared gradient satisfies

$$\lim_{\eta \to 0} \mathcal{L}_{\text{ECO}} = \lim_{\eta \to 0} L^2 \mathbb{E}[\hat{x}^2] = \frac{L^2 \sigma^2}{1 - \beta^2}. \tag{108}$$

**Interpretation.** Comparing (98) and (108), naive master-weight removal yields a stationary error that blows up like $1/\eta$, while ECO stabilizes the dynamics and yields a finite noise floor controlled by the geometric factor $1/(1 - \beta^2)$.

