# OpenReview forum: "ECO: Quantized Training without Full-Precision Master Weights"
_ICML.cc/2026/Conference — ICML 2026 regular_

### Official Review · Reviewer_hsnv · 2026-03-12

**Soundness:** 3
**Presentation:** 3
**Significance:** 3
**Originality:** 3
**Overall Recommendation:** 4
**Confidence:** 3

**Summary:**

This paper introduces the Error-Compensating Optimizer (ECO), a novel approach designed to eliminate the need for full-precision master weights in quantized Large Language Model (LLM) training. By quantizing the weights after each update step and injecting the resulting quantization error directly into the optimizer's momentum buffer, ECO creates an error-feedback loop that preserves minute gradient updates without requiring additional memory. The authors provide theoretical convergence guarantees for ECO under standard non-convex assumptions and demonstrate its empirical effectiveness across various settings, including pre-training small Transformers, Gemma-3 1B, a 2.1B SMoE model, and fine-tuning DeepSeek-MoE-16B.

**Compliance With Llm Reviewing Policy:**

Affirmed.

**Final Justification:**

This paper proposes a training-time master weight quantization technique that is highly compatible with optimizer state compression methods. Therefore, I lean towards accepting it.

**Key Questions For Authors:**

1. Compatibility with 8-bit Adam: Can ECO function correctly if the optimizer states (specifically the momentum buffer where errors are injected) are quantized to 8-bit?

2. Could the authors provide a direct empirical comparison (in terms of both peak memory footprint and validation loss) between:
(A) ECO with FP8 weights and FP32 Adam states (the proposed method).
(B) Standard training with FP32 master weights, FP8 forward weights, but using 8-bit Adam?
This comparison is crucial to demonstrate ECO's practical value in the current landscape of memory-reduction techniques.

**Limitations:**

Yes

**Strengths And Weaknesses:**

Strengths:
1. The core idea of repurposing the existing momentum buffer to store quantization errors (Error Feedback) is elegant. It effectively addresses a real-world bottleneck (the memory footprint of FP32 master weights) with zero additional memory overhead.
2. The paper provides comprehensive theoretical proofs.
3. The experimental results are convincing.

Weaknesses:
1. Lack of Discussion and Compatibility with Memory-Efficient Optimizers:
The most significant weakness of this paper is its narrow focus on standard, full-precision optimizers (like standard AdamW), completely ignoring the broader ecosystem of memory-efficient optimizers widely used in modern LLM training.

ECO's core mechanism fundamentally relies on the existence of a full-parameter, high-precision first-moment buffer to store the quantization error. This creates severe potential conflicts: a) Conflict with 8-bit Optimizers (e.g., 8-bit Adam): A highly popular method for reducing training memory is quantizing the optimizer states to 8-bit. If the momentum buffer is in 8-bit, the minute quantization errors injected by ECO will likely be truncated or lost during the quantization/dequantization; b) Conflict with Adafactor: Adafactor achieves massive memory savings by factoring the second moment and, by default, completely discarding the first momentum buffer.

2. Because the authors do not compare ECO with optimizer-state quantization techniques, the true Pareto frontier of memory vs. performance is incomplete

---

> ### Author Rebuttal · Authors · 2026-03-31
>
> We would like to thank the reviewer for their insightful comments.
>
> To study the interaction between ECO and optimizer state compression methods (namely COAT [1] and 8-bit Adam [2]), we pre-train the 30M and 50M models in FP8 row-wise format with a Chinchilla-optimal number of tokens (1/5 the duration of the paper’s scaling law experiments). For the purpose of this study, we keep the second Adam moment in high precision (as it has no direct interaction with ECO) and study different combinations of master weight and first moment quantization. Namely, we include the following baselines, and for each, we consider different momentum precisions:
> - FP8 w/ MW + RTN
> - FP8 w/ MW + SR
> - FP8 w/o MW + SR
> - FP8 w/o MW + SR + ECO
>
> The validation loss of each method is reported in the two tables below, one for each of 30M and 50M model sizes.
>
> | Validation Loss (30M) | FP32 Momentum | 8-bit Momentum (COAT) | 8-bit Momentum (8-bit Adam) |
> |---|---|---|---|
> | FP8 w/ MW + RTN | 3.4633 | 3.4632 | 3.4644 |
> | FP8 w/ MW + SR | 3.4673 | 3.4663 | 3.4690 |
> | FP8 w/o MW + SR | 3.5104 | 3.5075 | 3.5098 |
> | FP8 w/o MW + SR + ECO | 3.4705 | 3.4651 | 3.4684 |
>
> | Validation Loss (50M) | FP32 Momentum | 8-bit Momentum (COAT) | 8-bit Momentum (8-bit Adam) |
> |---|---|---|---|
> | FP8 w/ MW + RTN | 3.2811 | 3.2809 | 3.2813 |
> | FP8 w/ MW + SR | 3.2840 | 3.2831 | 3.2851 |
> | FP8 w/o MW + SR | 3.3311 | 3.3343 | 3.3333 |
> | FP8 w/o MW + SR + ECO | 3.2842 | 3.2856 | 3.2827 |
>
> Discussion on Compatibility: Interestingly, the results clearly indicate that ECO and Momentum compression are compatible in the 8-bit setting. This means ECO can be applied on top of existing optimizer state compression, achieving extra memory savings **for free**. We hope these results address the reviewers' concerns regarding ECO’s significance in the presence of optimizer state compression methods.
>
> Discussion on Memory: Below we compare the static memory requirement of each method.
>
> MW + FP32 optimizer: 4 bytes/param weight + 4 bytes/param first momentum + 4 bytes/param second momentum -> 12 bytes/param
>
> ECO + FP32 optimizer: 1 bytes/param weight + 4 bytes/param first momentum + 4 bytes/param second momentum -> 9 bytes/param (25% reduction)
>
> ECO + 8-bit first moment + FP32 second moment: 1 bytes/param weight + 1 bytes/param first momentum + 4 bytes/param second momentum -> 6 bytes/param (50% reduction)
>
> While we do not consider this case in our experiments due to the limited time of the rebuttal period, additionally compressing the second moment to 8-bit would achieve a 75% memory reduction compared to the baseline.
>
> In summary, this study shows that ECO is compatible with 8-bit optimizer state compression methods in an almost lossless way, which further shifts the static memory vs loss pareto frontier.
>
> We will include these studies in the next revision of the paper.
>
> [1] https://arxiv.org/pdf/2410.19313
> [2] https://arxiv.org/pdf/2110.02861

---

> > ### Author Rebuttal · Reviewer_hsnv · 2026-04-04
> >
> > My concern has been all resolved.

---

### Official Review · Reviewer_NFCx · 2026-03-12

**Soundness:** 4
**Presentation:** 3
**Significance:** 3
**Originality:** 3
**Overall Recommendation:** 5
**Confidence:** 4

**Summary:**

This paper proposes a memory-efficient LLM pretraining/finetuning scheme that allows the removal of high precision master weight for gradient accumulation. Specifically, ECO quantizes weights after each step and carefully injects the resulting quantization error into the optimizer momentum, forming an error-feedback loop with no additional memory. Theoretical justification is provided on the convergence guarantee.

**Compliance With Llm Reviewing Policy:**

Affirmed.

**Final Justification:**

This is a solid work. The rebuttal strengthens my confidence in supporting this work.

**Key Questions For Authors:**

Can the proposed method be combined with optimizer state quantization like COAT? How does the memory-loss tradeoff of the proposed method compare with optimizer state quantization based method?

**Limitations:**

Yes

**Strengths And Weaknesses:**

This paper looks at the important topic of training memory reduction from a novel len of master weight removal. The method appears to be simple but is well motivated and technically sound. Detailed theoretical derivation is provided. The derivation is clear and correct. Method is evaluated across multiple base model precision and model architecture, showing the effectiveness of the method.

One potential weakness is the lack of direct comparison against optimizer state quantization methods like COAT cited in Section 2. As I understand the different focus, it would be interesting to see if the two side can work together to lead to even better memory-loss tradeoff. If the two methods cannot be easily combined, then it would be interesting to see which track leads to more promising tradeoff so as to prove the significance of the proposed method.

---

> ### Author Rebuttal · Authors · 2026-03-31
>
> We would like to thank the reviewer for their insightful comments.
>
> To study the interaction between ECO and optimizer state compression methods (namely COAT [1] and 8-bit Adam [2]), we pre-train the 30M and 50M models in FP8 row-wise format with a Chinchilla-optimal number of tokens (1/5 the duration of the paper’s scaling law experiments). For the purpose of this study, we keep the second Adam moment in high precision (as it has no direct interaction with ECO) and study different combinations of master weight and first moment quantization. Namely, we include the following baselines, and for each, we consider different momentum precisions:
> - FP8 w/ MW + RTN
> - FP8 w/ MW + SR
> - FP8 w/o MW + SR
> - FP8 w/o MW + SR + ECO
>
> The validation loss of each method is reported in the two tables below, one for each of 30M and 50M model sizes.
>
> | Validation Loss (30M) | FP32 Momentum | 8-bit Momentum (COAT) | 8-bit Momentum (8-bit Adam) |
> |---|---|---|---|
> | FP8 w/ MW + RTN | 3.4633 | 3.4632 | 3.4644 |
> | FP8 w/ MW + SR | 3.4673 | 3.4663 | 3.4690 |
> | FP8 w/o MW + SR | 3.5104 | 3.5075 | 3.5098 |
> | FP8 w/o MW + SR + ECO | 3.4705 | 3.4651 | 3.4684 |
>
> | Validation Loss (50M) | FP32 Momentum | 8-bit Momentum (COAT) | 8-bit Momentum (8-bit Adam) |
> |---|---|---|---|
> | FP8 w/ MW + RTN | 3.2811 | 3.2809 | 3.2813 |
> | FP8 w/ MW + SR | 3.2840 | 3.2831 | 3.2851 |
> | FP8 w/o MW + SR | 3.3311 | 3.3343 | 3.3333 |
> | FP8 w/o MW + SR + ECO | 3.2842 | 3.2856 | 3.2827 |
>
> Discussion on Compatibility: Interestingly, the results clearly indicate that ECO and Momentum compression are compatible in the 8-bit setting. This means ECO can be applied on top of existing optimizer state compression, achieving extra memory savings **for free**. We hope these results address the reviewers' concerns regarding ECO’s significance in the presence of optimizer state compression methods.
>
> Discussion on Memory: Below we compare the static memory requirement of each method.
>
> MW + FP32 optimizer: 4 bytes/param weight + 4 bytes/param first momentum + 4 bytes/param second momentum -> 12 bytes/param
>
> ECO + FP32 optimizer: 1 bytes/param weight + 4 bytes/param first momentum + 4 bytes/param second momentum -> 9 bytes/param (25% reduction)
>
> ECO + 8-bit first moment + FP32 second moment: 1 bytes/param weight + 1 bytes/param first momentum + 4 bytes/param second momentum -> 6 bytes/param (50% reduction)
>
> While we do not consider this case in our experiments due to the limited time of the rebuttal period, additionally compressing the second moment to 8-bit would achieve a 75% memory reduction compared to the baseline.
>
> In summary, this study shows that ECO is compatible with 8-bit optimizer state compression methods in an almost lossless way, which further shifts the static memory vs loss pareto frontier.
>
> We will include these studies in the next revision of the paper.
>
> [1] https://arxiv.org/pdf/2410.19313
> [2] https://arxiv.org/pdf/2110.02861

---

> > ### Author Rebuttal · Reviewer_NFCx · 2026-04-01
> >
> > The results provided by the author in the rebuttal proves the compatibility of the proposed method and existing optimizer state compression method. To this end, my concern is resolved.

---

### Official Review · Reviewer_r44a · 2026-03-12

**Soundness:** 3
**Presentation:** 3
**Significance:** 2
**Originality:** 3
**Overall Recommendation:** 4
**Confidence:** 3

**Summary:**

The Error-Compensating Optimizer (ECO) tackles a specific use case that is memory-efficient quantized (and quantization-aware) training without the preservation of full-precision master weights, which are typically used to accumulate gradients for the quantized model copy. Especially for large models like LLMs this is restrictive or unfeasible. So ECO makes use of a quite elegant trick, that is injecting quantization error (i.e., residuals $e_{\theta} = \theta - q(\theta))$ into the optimizer’s momentum buffer, creating an error feedback loop that compensates for the reduced update fidelity caused by quantization in subsequent training steps.

**Compliance With Llm Reviewing Policy:**

Affirmed.

**Final Justification:**

In acknowledgement of the rebuttal, I have raised the soundness score from 2 (fair) to 3 (good) and the overall score from 3 (weak reject) to 4 (weak accept). Since I can only hope that the authors will include further discussion of the applicability to other neural architectures, tasks, and use cases, as well as provide more detail on implementation aspects (e.g., how readiy it can be integrated into other environments and under which conditions ECO remains stable), and they did not clearly indicate a willingness to incorporate such changes into the paper, I will keep the already positive scores and do not update them further.

**Key Questions For Authors:**

1) Only linear layers within transformer blocks are quantized. What relative amount of the net topologies is excluded here and what would be the effect if all layers were quantized (i.e., is it minor or would it cause serious harm to the model)?
2) Why does the w/o MW + SR method diverge in 4.5 (Figure 4) but in Table 1 achieve quite some good convergence? Is it only due to reduced precision or the different task or ...?
3) Why is stochastic rounding so hardware-bound (as described in the Limitations section)?
4) How would the method perform with other tasks (e.g., vision) and models (e.g., CNNs)?
5) Memory Analysis: Authors discuss "Reducing master weight precision from FP32 to FP8 therefore lowers peak memory consumption from 12 bytes per parameter to 9, a reduction of approximately 25%." However, eliminating master weights should increase memory savings of the overall approach even more? What are the maximum savings in memory when comparing the
- FP16/FP32 baseline with (compressed) master weights and uncompressed optimizer states
- FP16/FP32 baseline with (compressed) master weights and compressed optimizer states
- ECO with (different bitdepths) compressed optimizer states,

and how does it affect training convergence and final model accuracy?

**Limitations:**

yes

**Strengths And Weaknesses:**

Strengths:

- The work closes the following gap: it presents a general method that eliminates master weights from the QAT paradigm without sacrificing model convergence, thereby saving ~25% (and potentially more?) memory cost while training.
- Trick: reusing the optimizer’s momentum buffer to restore quantization error in subsequent training iterartions.
- For SGD with momentum (SGDM), an error injection is proposed that leads to an optimization trajectory identical to SGDM with master weights, but incurs high memory costs due to residual buffers from previous steps. The authors find a simplified heuristic that leads to an approximate, memory-free trajectory which is original and also applicable to the Adam optimizer. The theoretical foundations and assumptions for the construction of the method are well-founded.
- No runtime overhead.

Weaknesses:

- Major: Scope is a bit narrow / specialized, i.e., Quantized/Quantization-Aware Training (here only of LLMs).

- Major: Experimental design is not convincing. Table 1: what type of loss is reported? Loss of FP8 w/o MW + SR  fairly close to the ECO results (not plotted in Figure 1), so it would be interesting to see how these minor differences in loss translate to potential gains in better defined tasks / benchmarks that are comparable to related works. My problem with the current experimental section is that I am not convinced by the broader effectivity of the method. Additionally to Table 2, it would be good to see some validation / test performance plots showing re-training behaviour of models recovering from quantization.

- Minor: Panferov reference lacks publication year.

- Minor: Redundance: End of section 3.1 (Quantization-Aware Training with Master Weights) and first two paragraphs in 3.1 are somewhat redundant, this was already explained sufficiently. The experimental setting blocks are also quite redundant since they are put before each sub-experiments, however often using common settings.

- Minor: consistently use Table captions above or below the Table (preferably above).

---

> ### Author Rebuttal · Authors · 2026-03-31
>
> We thank the reviewer for the comments and address the concerns below.
>
> > 1. Scope limited to quantization.
>
> Although quantized training may seem specialized, it is now a standard method for scaling LLMs. FP8 training is now widely used, and the a similar QAT setup to the one we study is used in at least one open frontier model (Moonshot Kimi K2.5). Yet high-precision master weights remain necessary in practice, limiting the memory benefits of quantization. A rigorous, near-lossless solution to this overhead therefore has broad practical value.
>
> > 2. Questions about experimental design.
>
> We thank the reviewer for their detailed evaluation.
>
> **Reported loss**
>
> Table 1 reports standard next-token cross-entropy on a held-out C4 validation set, the standard metric for scaling-law studies [1][2][3].
>
> **Comparison with the naive w/o MW + SR baseline**
>
> This baseline already appears in Figure 1 (red dashed) and becomes unstable at larger scales. The gap to ECO is substantial. At 800M parameters, ECO reaches 2.5399 validation loss, versus 2.9471 for the naive baseline. In language modeling, a ~0.4 cross-entropy gap corresponds to a large perplexity reduction (about 19.0 to 12.6). The naive 800M model performs roughly in the range of the 100M ECO model, about 8x smaller.
>
> **Downstream Tasks**
>
> We agree downstream transfer matters, and Table 2 evaluates precisely this on DeepSeek-MoE-16B. We did not report the w/o MW + SR baseline there because it diverged during training. There, ECO is the difference between successful training and divergence.
>
> We also include a new study of bit-widths and group sizes; see rebuttal to Reviewer sJU3, point 3.
>
> [1] https://arxiv.org/pdf/2001.08361
> [2] https://arxiv.org/pdf/2502.05003
> [3] https://arxiv.org/pdf/2309.08520
>
> > 3. Missing publication year, Redundancy, Table caption misplacement
>
> We thank the reviewer for the careful reading and will incorporate these corrections in the next revision.
>
> > 4. Quantization of non-linear layers
>
> We thank the reviewer for raising this point. The excluded layers are normalization layers, embedding layers, and the LM head. This is standard in the QAT literature.
>
> If the model has L transformer blocks, hidden size H, and vocabulary size V, then linear layers contain O(LH^2) parameters, normalization layers O(LH), and embeddings/LM head O(VH). Thus normalization layers are negligible, and embeddings/LM head become a relatively small fraction as models grow.
>
> Quantizing embeddings and the LM head would likely hurt convergence and final quality. Leaving the first and last layers in higher precision is standard practice in LLM quantization [2][4][5], and common PTQ methods such as GPTQ [6] also quantize only linear layers. We therefore quantize the dominant bulk of parameters while keeping the most sensitive layers in high precision.
>
> We will clarify this in the experiments section.
>
> [4] https://arxiv.org/pdf/2402.17764
> [5] https://arxiv.org/pdf/2407.12327
> [6] https://arxiv.org/pdf/2210.17323
>
> > 5. Divergence of w/o MW + SR in the DeepSeek experiment
>
> Our new integer-quantization results (see rebuttal to Reviewer sJU3, point 3) show that even INT1 does not cause complete divergence for the small 30M model. However, Figure 1 already shows that w/o MW + SR becomes increasingly unstable as model size grows, even in FP8. We therefore suspect the Figure 4 divergence is mainly due to DeepSeek’s scale. In contrast, ECO remains stable across larger models and tasks.
>
> We will discuss this in the next revision.
>
> > 6. SR hardware limitation
>
> While SR is natively supported by the latest flagship accelerators (such as Nvidia H100), it lacks hardware support on older GPUs (such as A100s).
>
> > 7. Application of ECO to CNNs
>
> ECO’s theoretical foundation is architecture-agnostic, so the same method should apply to other architectures including CNNs
>
> We focus on LLMs and MoEs because in these models, master weights occupy a large fraction of training memory. In CNNs, activations usually dominate memory instead, so ECO would reduce weight memory but likely yield smaller system-level gains than in LLMs.
>
> We will include this discussion in the next revision.
>
> > 8. Memory Analysis
>
> Assuming activation checkpointing is enabled, the static memory will consist of the weights and two Adam momentums. Each of these will introduce 4 bytes/param -> 12 byte/param in total for the high precision baseline. ECO reduces the size of weights from 4 to 1 bytes (FP8), totaling 9 bytes/param. Therefore, ECO can achieve up to 25% memory savings in the FP8 setting.
>
> ECO alone can only remove the overhead of the weights, so 8/12 of the bytes are not affected. Hence the memory saving is bounded by 33.3%. Refer to the rebuttal to Reviewer sJU3 point #3 to see a study on the accuracy of different weight precisions.
>
> Another interesting direction is to study the interaction of ECO and optimizer state compression. Please see point #1 in the rebuttal to Reviewer NFCx for a discussion on this.

---

> > ### Author Rebuttal · Reviewer_r44a · 2026-04-01
> >
> > Thanks for the clarifications regarding loss and comparison with naive w/o MW + SR baseline. I still think it wouldn't hurt to mention that cross-entropy is reported (e.g., in the Setting paragraph or Table’s caption). You could even consider reporting perplexity alongside cross-entropy (e.g., in Table 1 *or* Figure 1). Although it's just another scale, it might improve comparability and intuitive understanding. Your rebuttal helped me better understand the paper. I suggest incorporating the clarifications to make the paper in general more understandable.
> >
> > I think the new investigations on different bit depths and the interaction between ECO and additional optimizer state compression are highly valuable. They increase the paper's maturity and enrich the experimental methodology. Therefore, I increased the soundness score from 2 (fair) to 3 (good) and the overall score from 3 (weak reject) to 4 (weak accept). Further explanations of the applicability to other neural architectures, tasks, and use cases, as well as implementation aspects (e.g., whether it is straightforward to implement in other environments and under which settings ECO performs stably) would further underscore the significance of the paper.

---

### Official Review · Reviewer_sJU3 · 2026-03-12

**Soundness:** 2
**Presentation:** 2
**Significance:** 2
**Originality:** 3
**Overall Recommendation:** 4
**Confidence:** 4

**Summary:**

This paper proposes ECO, a quantized training framework that removes the need for full-precision master weights. The key idea is to inject quantization errors into the optimizer update so that weight updates can be performed directly on quantized weights. The method is motivated by the observation that consecutive quantization errors are similar, which allows the algorithm to approximate the exact update rule without storing previous residuals. The paper provides theoretical analysis under standard assumptions and empirical evaluations on several language models and sparse MoE models.

The problem addressed is important, as master weights introduce substantial memory overhead in large-scale training, especially for sparse MoE models. However, several aspects of the method and evaluation raise concerns about the robustness of the assumptions and the generality of the empirical results.

**Compliance With Llm Reviewing Policy:**

Affirmed.

**Final Justification:**

My main concerns have been adequately resolved, and I maintain a positive overall assessment of the paper.

**Key Questions For Authors:**

1. How sensitive is ECO to different quantization methods and rounding strategies?
2. Can the authors provide experiments on more recent large-scale architectures or more challenging MoE configurations?
3. Is it possible to quantify the error introduced by the \(e_t \approx e_{t+1}\) approximation in the optimizer update?

**Limitations:**

yes

**Strengths And Weaknesses:**

### Strengths

- **Practical motivation.** Removing full-precision master weights is an important problem for large-scale quantized training, particularly in memory-constrained settings such as sparse MoE training.
- **Simple and implementable approach.** ECO introduces a relatively lightweight modification to the optimizer update that can be integrated into existing training pipelines.
- **Initial empirical evidence.** The paper demonstrates that ECO can maintain stable training in several models and quantization settings, suggesting the idea is promising.

---

### Weaknesses

**1. The core approximation may introduce non-negligible error.**

The proposed algorithm relies on the assumption that consecutive quantization errors \(e_t\) and \(e_{t+1}\) are similar, allowing the update rule to replace \(e_t\) with \(e_{t+1}\) and avoid storing residuals. However, Figure 2 suggests that the cosine similarity between consecutive errors can drop significantly during early training. This raises concerns about the accuracy of the approximation, especially during the high-learning-rate phase when training is most sensitive. The theoretical analysis does not appear to explicitly bound the error introduced by this approximation.

---

**2. Experimental evaluation may not reflect modern large-scale models.**

The experiments include models such as DeepSeek-MoE-16B and smaller dense models. While these are reasonable benchmarks, they are relatively dated compared with recent architectures with larger expert counts and smaller activation ratios (e.g., newer MoE families). Since the paper emphasizes scalability and applicability to modern LLM training, stronger evidence on more recent architectures would strengthen the claims.

---

**3. Sensitivity to quantization schemes is not sufficiently explored.**

The theoretical guarantees assume unbiased quantization noise, which holds primarily for stochastic rounding. However, in practice different quantization schemes (e.g., deterministic rounding, outlier-aware quantization, different group sizes) may produce significantly different error distributions. Since ECO directly relies on quantization error injection, its effectiveness may depend strongly on the quantization method used. The paper would benefit from a more systematic study of how ECO behaves under different quantization schemes.

---

**4. Assumptions in the theoretical analysis appear strong.**

The convergence analysis relies on standard assumptions such as unbiased quantization noise and bounded gradients. While common in optimization theory, it is unclear how well these assumptions hold in large-scale LLM training with quantized weights. The gap between the theoretical setting and the practical training regime could be discussed more carefully.

---

> ### Author Rebuttal · Authors · 2026-03-31
>
> We address the reviewer's constructive comments below.
>
> > 1. Discussion on the $e_t \approx e_{t+1}$ approximation
>
> We thank the reviewer for the observation. You are correct that the cosine similarity between consecutive errors drops during the early phase of training. However, our theoretical analysis **does not** rely on the $e_t \approx e_{t+1}$ approximation. As mentioned in Section 3.2 (Line 182), this substitution is used purely as an intuitive heuristic to motivate the memory-free injection rule.
>
> Our theoretical framework does not quantify the error introduced by this approximation because it does not need to. Our theoretical results analyze the actual memory-free ECO algorithm directly, exactly as it is implemented. The analysis proves that this standalone injection rule converges to a tight neighborhood of the optimum, independent of whether $e_t$ matches $e_{t+1}$.
>
> We hope this clarifies the point.
>
> > 2. Experiments on more recent/challenging MoEs
>
> We agree that evaluating on even larger and newer models would be valuable. However, we would like to clarify that the models in our current evaluation already cover recent architectures and challenging sparse MoE settings:
>
> - We pre-train Gemma 3 1B, Google’s state-of-the-art open model released in 2025.
> - We pre-train a highly sparse SMoE 2.1B model, where only 4 of 32 experts are active.
> - We fine-tune DeepSeek-MoE-16B, which is also highly sparse, with 8 of 64 experts active, including 2 shared experts.
>
> These settings already reflect modern sparse regimes with low activation ratios. In addition, our theoretical results are model-agnostic and do not depend on any specific model family. In total, we used about 5000 H100 GPU hours for our current set of experiments.
>
> That said, training a new, larger-scale MoE during the rebuttal period is unfortunately beyond our computational budget, and we believe should not be a prerequisite for acceptance to ICML. We therefore hope the combination of (1) general theoretical guarantees and (2) strong empirical results on recent and challenging architectures, including Gemma 3 1B, SMoE 2.1B, and DeepSeek-MoE-16B, provide convincing support for ECO's generalizability.
>
> > 3.Study on the quantization schemes.
>
> We thank the reviewer for this valuable comment. To provide a systematic study on quantization schemes within the time limits of the rebuttal, we pre-train the 30M model with a Chinchilla-optimal number of tokens. We try weight-only INT8, INT4, INT3, INT2, and INT1 precisions with QuEST’s MSE-optimal scaling [1]. We tested row-wise quantization and group sizes of 128 and 32.
>
> |Row-wise|INT8|INT4|INT3|INT2|INT1|
> |---|---|---|---|---|---|
> |w/ MW+RTN|3.4636|3.4944|3.5260|3.5985|3.9525|
> |w/ MW+SR|3.4659|3.5183|3.5964|3.7323|4.0927|
> |w/o MW+SR|3.5309|3.8255|3.9840|4.1158|5.1324|
> |w/o MW+SR+ECO|3.4649|3.5445|3.6417|3.8273|4.7508|
>
> |GroupSize128|INT8|INT4|INT3|INT2|INT1|
> |---|---|---|---|---|---|
> |w/ MW+RTN|3.4623|3.4897|3.5237|3.6084|3.9594|
> |w/ MW+SR|3.4645|3.5193|3.5934|3.7289|4.1138|
> |w/o MW+SR|3.5293|3.9420|4.2001|4.2232|5.1101|
> |w/o MW+SR+ECO|3.4661|3.5514|3.6538|3.8673|4.7696|
>
> |GroupSize32|INT8|INT4|INT3|INT2|INT1|
> |---|---|---|---|---|---|
> |w/ MW+RTN|3.4626|3.4892|3.5290|3.6009|3.9690|
> |w/ MW+SR|3.4641|3.5126|3.5887|3.7317|4.1343|
> |w/o MW+SR|3.5330|4.3067|4.6982|4.2519|5.1225|
> |w/o MW+SR+ECO|3.4630|3.5684|3.6942|3.9855|4.7534|
>
> - Consistent with our FP8 experiments, ECO successfully recovers the master weight baselines at INT8, establishing it as the first method to achieve near-lossless results in 8-bit precision without master weights.
> - As precision decreases, ECO consistently and significantly outperforms the naive approach, though the gap to the master-weight baseline naturally increases due to extreme quantization noise.
> - Group size does not have a significant impact on the results. We attribute this stability to the MSE-optimal scaling and the near-Gaussian distribution of the weights.
>
> Regarding the rounding, as supported by our theoretical bounds, ECO outperforms the naive approach when using deterministic RTN. However, because SR maintains the unbiased noise assumption, ECO achieves its tightest convergence and best empirical results when paired with SR.
>
> We thank the reviewer and will include these results in the next revision of the paper.
>
> [1] https://arxiv.org/pdf/2502.05003
>
> > 4. Discussion on theoretical assumptions
>
> We would like to clarify two points:
>
> - When using SR, the unbiased quantization noise assumption is strictly satisfied. In addition to that, in Theorem 3.10, we explicitly provide a convergence bound for RTN, where the unbiased noise assumption is violated.
>
> - While it is true that the loss landscape can produce gradient spikes, practical LLM training settings employ gradient clipping, which by definition bounds the update norm. Notably, as mentioned in the paper, our experiments also include gradient clipping.
>
> We will discuss these points more clearly in the next revision.

---

> > ### Author Rebuttal · Reviewer_sJU3 · 2026-04-02
> >
> > I appreciate the authors’ detailed rebuttal. After reviewing their response, I am keeping my overall recommendation as Weak Accept.

---

### Decision · Program_Chairs · 2026-04-30

**Decision:**

Accept (regular)

**Comment:**

While quantization-aware training has effectively reduced the memory footprint of forward weights, standard paradigms still necessitate maintaining high-precision master weights to accumulate minute gradient updates. The reviewer consensus is positive, with scores settling at Weak Accept (x3) and Accept (x1) following a highly effective rebuttal period. The committee finds this work to be a timely, theoretically sound, and highly practical contribution to the deep learning systems community.